# PERFGUARD: A PERFORMANCE-AWARE AGENT FOR VISUAL CONTENT GENERATION

**Zhipeng Chen**[1,2], **Zhongrui Zhang**[1], **Chao Zhang**[2,*], **Yifan Xu**[3], **Lan Yang**[1,4], **Jun Liu**[1,5], **Ke Li**[1,4,5,*], and **Yi-Zhe Song**[4]

[1]School of Artificial Intelligence, Beijing University of Posts and Telecommunications, China
[2]Beijing Digital Native Digital City Research Center, China
[3]School of Computer Science and Engineering, Beihang University, China
[4]SketchX, CVSSP, University of Surrey, United Kingdom
[5]Chenxi Shuzhi (Beijing) Technology Co., Ltd, China
`{zhipengchen1998, zhongrui_zhang, ylan, liujun, like1990}@bupt.edu.cn, ariczhang2009@gmail.com, yifan_xu@buaa.edu.cn, y.song@surrey.ac.uk`

## ABSTRACT

The advancement of Large Language Model (LLM)-powered agents has enabled automated task processing through reasoning and tool invocation capabilities. However, existing frameworks often operate under the idealized assumption that tool executions are invariably successful, relying solely on textual descriptions that fail to distinguish precise performance boundaries and cannot adapt to iterative tool updates. This gap introduces uncertainty in planning and execution, particularly in domains like visual content generation (AIGC), where nuanced tool performance significantly impacts outcomes. To address this, we propose PerfGuard, a performance-aware agent framework for visual content generation that systematically models tool performance boundaries and integrates them into task planning and scheduling. Our framework introduces three core mechanisms: (1) Performance-Aware Selection Modeling (PASM), which replaces generic tool descriptions with a multi-dimensional scoring system based on fine-grained performance evaluations; (2) Adaptive Preference Update (APU), which dynamically optimizes tool selection by comparing theoretical rankings with actual execution rankings; and (3) Capability-Aligned Planning Optimization (CAPO), which guides the planner to generate subtasks aligned with performance-aware strategies. Experimental comparisons against state-of-the-art methods demonstrate PerfGuard's advantages in tool selection accuracy, execution reliability, and alignment with user intent, validating its robustness and practical utility for complex AIGC tasks. The project code is available at `https://github.com/FelixChan9527/PerfGuard`.

## 1 INTRODUCTION

In recent years, with the continuous advancement of Large Language Model (LLM) technology (Guo et al., 2025; Yang et al., 2025a; Fang et al., 2025b), agent-based automated task processing has become an important research direction across various fields (Curtarolo et al., 2012; Gao et al., 2024b; Wang et al., 2024b; Agashe et al., 2025). By constructing system frameworks with logical reasoning capabilities and equipping agents with the ability to invoke external tools, researchers aim to achieve the decomposition, reasoning, and autonomous execution of complex tasks, thereby surpassing the limitations of traditional single tools or rule-based systems. Most existing research focuses on the task planning and tool scheduling strategies of agents, emphasizing the rationality of the planning process (Agashe et al., 2025; Zhang et al., 2025a; Hong et al., 2024a). However, these

---

\* Chao Zhang and Ke Li are co-corresponding authors: ariczhang2009@gmail.com, like1990@bupt.edu.cn

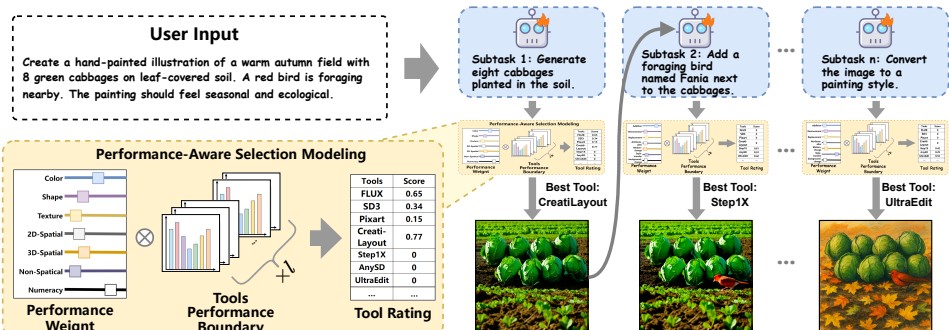

Figure 1: PerfGuard decomposes user requests into subtasks for iterative visual content generation. By modeling tool performance boundaries via PASM, it selects the most suitable tool in each round to ensure precise alignment between planning, execution, and user intent.

studies generally operate under the ideal assumption that "tool invocations are always successful," lacking systematic evaluation of the actual success rate of tool execution. Against this backdrop, how tool selection and their actual execution outcomes impact the overall accuracy of agent planning and decision-making remains a critical issue that has not been fully explored.

In current research, the description of tool capabilities often relies on general textual descriptions, which are difficult to accurately reflect their true performance boundaries. This issue is particularly prominent in the field of visual content generation (AIGC). Although existing systems (such as CompAgent (Wang et al., 2024c), GenArtist (Wang et al., 2024b), etc.) can enhance generation outcomes through task decomposition and multi-model scheduling, their descriptions of tool capabilities remain relatively coarse. These descriptions fail to clearly distinguish the specialized capabilities and applicable scenarios of different tools. Taking text-to-image generation as an example, common tool descriptions such as "capable of generating images aligned with the semantics of the input text" neither reflect the performance differences between various models nor support precise tool matching by agents in complex tasks, thereby introducing uncertainty into the planning and execution processes .

To address the aforementioned challenges, this paper introduces PerfGuard, a performance-aware agent framework for visual content generation. The framework aims to explore methods for modeling tool performance boundaries and leverage their impact on task planning and scheduling mechanisms. In response to the limitations posed by ambiguous tool capability descriptions, we propose Performance-Aware Selection Modeling (PASM), which replaces traditional textual descriptions with a multi-dimensional scoring mechanism based on fine-grained performance evaluation. Within this mechanism, the Worker dynamically selects the tool that best meets the performance requirements of the subtask generated by the Planner, thereby enhancing the accuracy and efficiency of task execution at the underlying scheduling level.

Acknowledging that preset performance boundaries (often derived from benchmark test results) may deviate from actual task execution outcomes, we further introduce an Adaptive Preference Updating (APU) method. This method continuously optimizes the performance boundary matrix by comparing the theoretical ranking of candidate tools with their observed performance during real task execution. This improves the accuracy of task-tool matching and enhances the system's adaptability to real-world scenarios.

To better align the task planning process with tool performance, we propose a Capability-Aligned Planning Optimization (CAPO) mechanism. This enables the Planner to generate high-quality task plans under the guidance of the performance-driven selection strategy facilitated by PASM. In each planning iteration, the Planner generates multiple candidate subtask plans and improves planning accuracy by comparing their output results. Through step-by-step supervision, the Planner learns to form planning patterns consistent with the performance-aware strategy, thereby systematically enhancing the robustness of the reasoning process .

To validate the effectiveness of PerfGuard, we conducted comparative experiments with existing representative visual content generation methods. In various tasks such as image generation and

editing, PerfGuard demonstrated advantages in tool selection accuracy, task execution reliability, and alignment with user intent. The results confirm the robustness and practical value of our framework.

## 2 RELATIVE WORKS

Recent advances in visual content generation have significantly improved controllability and semantic alignment. Models like FLUX (Labs, 2024), Stable Diffusion3 (Esser et al., 2024), and DALL·E3 (Betker et al., 2023) generate images from textual prompts, while ControlNet (Zhang et al., 2023), T2I-Adapter (Mou et al., 2024), InstanceDiffusion (Wang et al., 2024a) and VersaGen (Chen et al., 2025b) incorporate multimodal signals to better match user intent. To support fine-grained control, LayoutGPT (Feng et al., 2023), RPG (Yang et al., 2024b), GoT (Fang et al., 2025a), and T2I-R1 (Jiang et al., 2025) leverage LLMs to decompose prompts into region-specific semantics. Systems like CompAgent (Wang et al., 2024c) and GenArtist (Wang et al., 2024b) coordinate generation and editing tools, while MCCD (Li et al., 2025) and T2I-Copilot (Chen et al., 2025a) improve performance through model cooperation. CLOVA (Gao et al., 2024a) improves the success rate of visual tasks, including face swapping, by enhancing the tool's prompt pool through the introduction of self-reflection and prompt tuning. However, most approaches assume reliable tool execution and overlook how performance boundaries affect planning accuracy. PerfGuard addresses this gap by explicitly modeling tool capabilities and execution feedback.

## 3 PRELIMINARY

**Standardized Agent System**   Standardized agent systems typically consist of four core roles (Agashe et al., 2025; Hong et al., 2024b): Analyst, Planner, Worker, and Self-Evaluator. These roles handle task interpretation, planning, execution, and feedback respectively, enabling stepwise execution with continuous refinement. Building on this architecture, PerfGuard introduces Performance-Aware Selection Modeling to optimize tool selection for the Worker, ensuring better alignment with task requirements and improved execution performance.

**Step-aware Preference Optimization (SPO)**   In the visual domain, aligning image generation outputs with human aesthetic preferences has been a key challenge. Inspired by Direct Preference Optimization (DPO) (Rafailov et al., 2023) for aligning language model outputs with human preferences, researchers proposed Diffusion-DPO (Wallace et al., 2024) and D3PO (Yang et al., 2024a), which utilize a trained reward model to evaluate multiple random samples from a diffusion model and identify winning samples $x^w$ and losing samples $x^l$. To further improve the aesthetic quality of each intermediate step in the diffusion process, SPO (Liang et al., 2024) introduces a Step-Aware Preference Model (SPM) that evaluates and optimizes intermediate outputs at every step, ensuring that candidate samples are aligned with the optimal sample. The optimization objective is defined as:

$$\mathcal{L}(\theta) = - \mathbb{E}_{x_t^w, x_t^l \sim p_\theta(x_i|x_{t+1}, c, t+1)}$$
$$\left[ \log \sigma \left( \alpha \left( \log \frac{p_\theta(x_t^w \mid x_{t+1}^w, c, t+1)}{p_{\text{ref}}(x_t^w \mid x_{t+1}^w, c, t+1)} - \log \frac{p_\theta(x_t^l \mid x_{t+1}^l, c, t+1)}{p_{\text{ref}}(x_t^l \mid x_{t+1}^l, c, t+1)} \right) \right) \right] \tag{1}$$

where $\sigma$ denotes the sigmoid function, $c$ represents the input condition, $\alpha$ is a regularization hyper-parameter, $p_{\text{ref}}$ refers to the reference probability from the fixed initial denoising model $p_\theta$, and $\theta$ denotes the model parameters to be updated.

Motivated by SPO, we extend its methodology and apply the principle of stepwise intermediate output optimization to better align the Planner's decision-making and tool execution with optimal performance.

## 4 METHODOLOGY

Within the PerfGuard framework, we build on the standardized agent system to enable structured, stepwise planning and execution of visual generation tasks, as illustrated in Fig. 2. (1) Upon receiv-

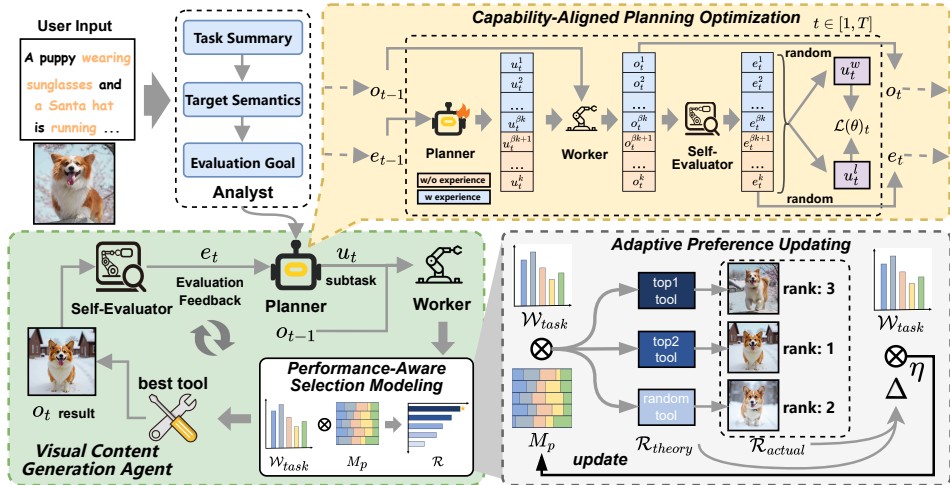

Figure 2: PerfGuard models tool performance boundaries via PASM to match the most suitable tool for each subtask, maximizing decision efficiency. It further integrates Adaptive Preference Updating to enhance real-world adaptability, and applies CAPO to align planning with performance-aware strategies.

ing multimodal inputs such as images or textual instructions, the Analyst parses the information to produce a task summary $\tau^*$, target image semantics $s^*$, and evaluation goals $g$. (2) The Planner uses $\tau^*$, $s^*$, and tool performance profiles $\mathcal{B}$ to decompose the task into subtasks $u_t$, which are executed by the Worker. Evaluation results $e_t$ from each stage are fed back to guide subsequent decisions $u_{t+1}$, enabling iterative refinement. (3) The Worker selects appropriate tools from the library to execute each $u_t$ and generate image outputs $o_t$. (4) Each output $o_t$ is assessed by the Self-Evaluator across multiple visual dimensions to measure alignment with goals $g$, providing feedback for continuous improvement. The definitions of agent roles and the tool library are provided in Appendix A.

## 4.1 PERFORMANCE-AWARE SELECTION MODELING

To rigorously define fine-grained tool performance boundaries, we propose the Performance-Aware Selection Modeling strategy. This method systematically aligns the Planner's subtasks with the most appropriate tools according to user-specified capability preference dimensions, thereby mitigating planning errors arising from ambiguous definitions of tool capabilities.

**Tool Performance Boundaries** Precise performance-aware scheduling begins with fine-grained performance boundary definition. We construct a multi-dimensional scoring system to evaluate tools in the library. Specifically, we design the performance boundary dimensions of tools by referring to authoritative benchmarks in image generation and editing. For image generation tools, semantic accuracy is assessed across seven dimensions including color, shape, texture, 2D spatial, 3D spatial, non-spatial semantics, and numeracy, based on T2I-compbench (Huang et al., 2023). For image editing tools, effectiveness is evaluated across seven dimensions including addition, removal, replacement, attribute alteration, motion change, style transfer, and background change, following the evaluation criteria defined in ImgEdit-Bench (Ye et al., 2025).

This multi-dimensional scoring framework enables flexible modeling across domains using standardized metrics from large-scale datasets to ensure fairness and objectivity. It supports accurate performance profiling and evolves with new tools and benchmarks. To reduce evaluation costs, we directly adopt scores from T2I-compbench and ImgEdit-Bench as the performance boundary matrices for generation and editing tools. A detailed description of the performance boundary dimensions and their design rationale is provided in Appendix A.7.

**Performance-Driven Selection** The Worker $\pi_{\text{Worker}}$ leverages predefined tool performance boundary dimensions $\mathcal{D}$ to select the most suitable tool for a sub-task $u_t$ provided by the Planner. For each $u_t$, the Worker leverages tool performance profiles to generate a preference weight $\mathcal{W}_{task} \in \mathbb{R}^{1 \times d}$, where $d$ denotes the number of performance dimensions. This vector captures the

relative importance of each dimension according to the characteristics of $u_t$. Task suitability scores $S_{tools}$ for all tools are then computed by combining $\mathcal{W}_{task}$ with the tool performance boundary matrix $M_p \in \mathbb{R}^{d \times l}$ (where $l$ tools have similar functionalities), enabling performance-driven tool selection. Formally, the computation is expressed as:

$$
\begin{aligned}
\mathcal{W}_{task} &= \pi_{\text{Worker}}(u_t, \mathcal{B}, \mathcal{D}) \\
S_{tools} &= \mathcal{W}_{task} \cdot \text{Normalize}(M_p)^\top \\
\mathcal{R} &= \text{argsort}(S_{tools}, \text{descending})
\end{aligned}
\tag{2}
$$

Here, Normalize$(\cdot)$ normalizes tool scores across all tools for each performance dimension. $S_{tools} \in \mathbb{R}^{1 \times l}$ represents the weighted suitability of all tools for $u_t$, and $\mathcal{R}$ provides their descending ranking. $\mathcal{B}$ denotes the information of the tool library This approach allows the system to automatically select tools based on their intrinsic performance characteristics, without requiring users to define task-specific preferences.

## 4.2 Adaptive Preference Updating

In practice, tool performance boundaries may originate from benchmarks or expert-like evaluations based on prior tool usage. These boundaries can contain inaccuracies due to differences in task-relevant dimensions or subjective biases. To enhance the accuracy of tool performance boundary scores, we propose an Adaptive Preference Updating mechanism that iteratively adjusts the scores based on actual tool usage. Specifically, during candidate tool selection, we implement an exploration-exploitation strategy: the top $m$ tools with the highest weighted preference scores are selected from the library, while $n$ additional tools are randomly sampled from the remaining ones to increase the likelihood of selecting potentially high-performing tools. This mechanism ensures that the tool performance boundary matrix $M_p$ more accurately reflects actual task requirements, enabling adaptive iterative updates:

$$
\begin{aligned}
\mathcal{R}_{theory} &= \text{top}_m(S_{tools}) \ \cup \ \text{rand}_n(S_{tools}[m+1:l]) \\
M_p^{\text{new}} &= \text{Normalize}\big(M_p + \mathcal{W}_{task} \cdot \eta \cdot \Delta\big) \\
\Delta &= \frac{\mathcal{R}_{theory} - \mathcal{R}_{actual}}{m + n}
\end{aligned}
\tag{3}
$$

Here, $\Delta$ represents the direction coefficient, reflecting the difference between the theoretical ranking $\mathcal{R}_{\text{theory}}$ and the actual usage ranking $\mathcal{R}_{\text{actual}}$, and $\eta$ denotes the update step size. When a tool's actual usage rank surpasses its theoretical rank, its performance boundary score is increased according to the weighted preferences and the distribution of task-specific emphasis across dimensions; otherwise, it is decreased. $\mathcal{R}_{\text{actual}}$ is derived from comparative evaluations of multiple candidate outputs conducted by a multimodal large model, with the evaluation procedure detailed in Appendix A.8. For newly added tools lacking sufficient usage experience or benchmark results, we initialize their scores using the average performance boundary scores of similar tools in the corresponding dimensions within the current library, ensuring that their potential is not overlooked in subsequent tool usage and iterative updates.

## 4.3 Capability-Aligned Planning Optimization

To further enhance the Planner's stepwise decision-making and provide indirect feedback on the execution effectiveness of tools selected via Performance-Aware Selection Modeling in PerfGuard, we extend Step-aware Preference Optimization (SPO) Liang et al. (2024) and propose Capability-Aligned Planning Optimization (CAPO) for the Planner's autoregressive planning process.

**Decision Performance Estimator** To evaluate the effectiveness of the Planner's output at each step $t$, we adopt the Self-Evaluator $\pi_{\text{Evaluator}}$ as the Planner's Decision Performance Estimator. For each sub-task execution result $o_t$, the Self-Evaluator assesses it based on the corresponding evaluation goals $g$ across multiple semantic dimensions:

$$
e_t = \sum_{i=0}^{L} \gamma_i^{local} \pi_{\text{Evaluator}}(o_t, g_i^{local}) + \gamma^{global} \pi_{\text{Evaluator}}(o_t, g^{global})
\tag{4}
$$

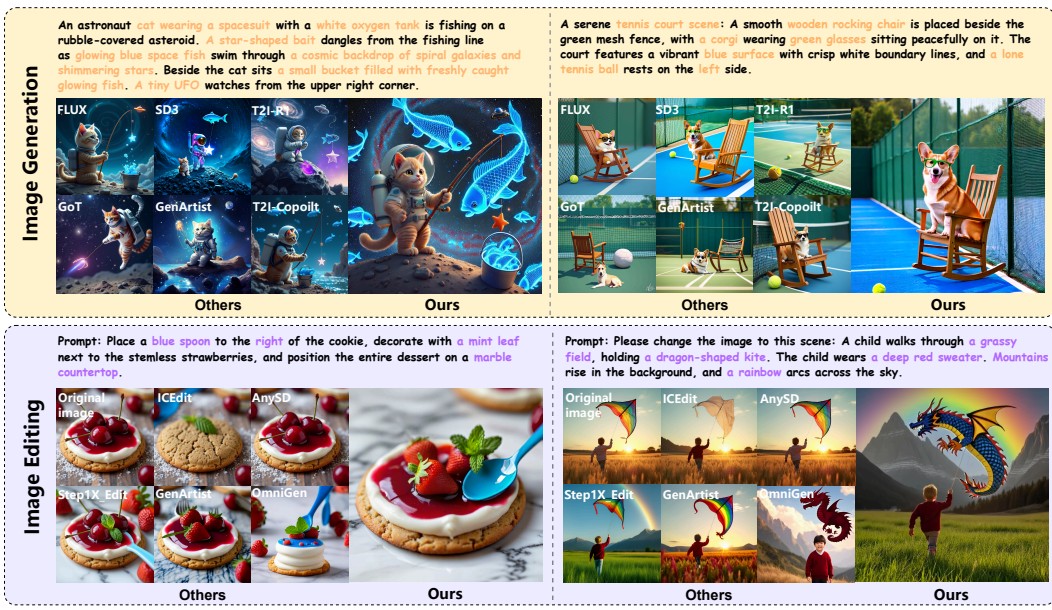

Figure 3: Comparison of PerfGuard's visual results across tasks. Top: visualization for complex text-to-image generation. Bottom: visualization for multi-round image editing.

Here, the evaluation goals consist of global semantics $g^{global}$ and local semantics $g_i^{local}$, weighted by $\gamma$. $L$ is the number of local dimensions, and $e_t$ denotes the Planner's decision evaluation at step $t$.

**Stepwise Planning Optimization**   At each step $t$, the Planner generates $k$ candidate sub-tasks $\{u_t^1, u_t^2, \ldots, u_t^k\}$. Each sub-task produces a corresponding output $\{o_t^1, o_t^2, \ldots, o_t^k\}$, which is evaluated by the Self-Evaluator. The sub-task with the highest evaluation score is selected as the winning sample $u_t^w$, and the lowest-scoring sub-task as the losing sample $u_t^l$. Accordingly, the planner's optimization objective function can be changed from Eq. 1 to:

$$\mathcal{L}(\theta) = - \mathbb{E}_{t \sim \mathcal{U}[1,T],\ u_t^w, u_t^l \sim \pi_{\text{Planner}}(\tau^*, s^*, \mathcal{B}, h_{t-1})}$$
$$\left[ \log \sigma \left( \alpha \left( \log \frac{p_\theta(u_t^w \mid \tau^*, s^*, \mathcal{B}, h_{t-1})}{p_{\text{ref}}(u_t^w \mid \tau^*, s^*, \mathcal{B}, h_{t-1})} - \log \frac{p_\theta(u_t^l \mid \tau^*, s^*, \mathcal{B}, h_{t-1})}{p_{\text{ref}}(u_t^l \mid \tau^*, s^*, \mathcal{B}, h_{t-1})} \right) \right) \right] \quad (5)$$

here, $h_{t-1} = \{(u_0, e_0), \ldots, (u_{t-1}, e_{t-1})\}$ denotes the history of sub-task executions and corresponding outputs evaluations up to timestep $t-1$.

CAPO enables the Planner to iteratively align sub-task decisions with feedback from the Self-Evaluator, enhancing its awareness of tool execution performance and thereby supporting more accurate and effective task planning.

To improve efficiency in trajectory data collection, a memory retrieval mechanism is integrated. Optimal sub-task sequences from previously successful tasks are stored as reusable experiences. During the generation of new candidate sub-tasks, an exploration–exploitation strategy is applied: among $k$ candidates, $\beta k$ are retrieved using CLIP Radford et al. (2021) similarity scores with the current task as the query, selecting the top-5 most similar sequences as contextual guidance, while the remaining $(1 - \beta)k$ candidates are generated randomly by the Planner.

## 5   EXPERIMENTS

We conducted both qualitative and quantitative comparisons of PerfGuard against various image generation and editing models. The evaluation spans three benchmarks covering different task types: basic image generation (T2I-CompBench (Huang et al., 2023)), advanced image generation (OneIG-Bench (Chang et al., 2025)), and complex image editing (Complex-Edit (Yang et al., 2025b)). De-

tailed experimental settings, descriptions of baseline methods, agent prompts and instructions, as well as additional results and visualizations, are provided in the supplementary material A.11..

## 5.1 QUALITATIVE RESULTS AND ANALYSIS

We compared the proposed PerfGuard with several existing methods on text-to-image generation and image editing tasks. The visualization results reveal three key observations: i) In text-to-image generation, traditional diffusion models struggle with complex prompts involving multiple entities and detailed attributes. Their limited language understanding leads to poor semantic alignment. For example, FLUX (Labs, 2024) and SD3 (Esser et al., 2024) fail to generate a cat in a space-suit. CoT-based methods like T2I-R1 (Jiang et al., 2025) and GoT (Fang et al., 2025a) incorporate LLMs, but due to reliance on a single-generation tool, they still miss key elements or actions, such as GoT omitting the fishing pose and several specified objects. Agent-based methods show improvement in semantic parsing and tool orchestration. However, GenArtist (Wang et al., 2024b) lacks a performance-aware tool selection strategy, resulting in planning errors and missing elements. T2I-Copilot (Chen et al., 2025a) performs better through multi-module semantic decomposition, but its limited tool diversity still leads to omissions, such as spiral galaxies and green glasses. ii) In multi-round editing tasks, traditional methods like ICEdit (Zhang et al., 2025c) and AnySD (Yu et al., 2025) deliver the weakest results. GenArtist, despite using multiple tools, suffers from poor capability matching, leading to suboptimal edits. Step1X_Edit (Liu et al., 2025) benefits from LLM-enhanced understanding of long instructions, but without intelligent planning and execution, it fails to capture key details—for example, the kite does not transform into a dragon. iii) Across both generation and editing tasks, PerfGuard consistently achieves the most accurate and visually aligned outputs. This demonstrates that its performance-guided tool selection enhances single-step execution accuracy and improves overall task planning.

## 5.2 QUANTITATIVE RESULTS AND ANALYSIS

To comprehensively validate the effectiveness of PerfGuard, we utilize three distinct benchmarks, namely T2I-CompBench (Huang et al., 2023), OneIG-Bench (Chang et al., 2025), and Complex-Edit (Yang et al., 2025b), to objectively evaluate its visual reasoning performance across both image generation and editing tasks from multiple perspectives.

**Basic Image Generation Comparison** We compare the proposed PerfGuard with various image generation methods on basic tasks, as shown in Tab 1. T2I-CompBench evaluates images in terms of attribute binding and object relationships. From the table: (i) Traditional models like FLUX and SD3 remain competitive, with texture, non-spatial, and complexity metrics approaching or surpassing CoT-based methods (T2I-R1, GoT). (ii) CoT-based methods rely on LLM fine-tuning, limiting them to certain tasks; simple prompts may yield overly complex interpretations and inaccurate images. (iii) Agent-based methods (GenArtist, T2I-Copilot) use self-correction to regenerate low-quality outputs, improving reliability. (iv) PerfGuard adapts capabilities to match the best-suited model for different tasks, achieving optimal performance across all dimensions.

**Advanced Image Generation Comparison** To further assess the effectiveness of our proposed method in visual reasoning, we evaluated various approaches on OneIG-Bench across diverse scenarios and complex text prompts, as shown in Tab. 2. (i) For more complex generation tasks, FLUX and SD3 show notably lower performance on reasoning metrics, highlighting that integrating LLMs improves the ability to handle complex information. (ii) Regarding alignment accuracy, GoT and GenArtist perform worse than other methods, indicating that a single large model has limited capacity for complex tasks. (iii) T2I-Copilot and PerfGuard (Ours), leveraging multi-agent collaboration, can plan each step of visual reasoning more precisely when handling cross-domain information, achieving optimal results in both alignment and reasoning metrics. (iv) PerfGuard does not show a large margin over other methods in alignment and text metrics due to toolset limitations, which cap its generation capabilities. However, its performance-aware tool selection enables smarter planning, leading to clear advantages in reasoning.

**Complex Image Editing Comparison** We evaluated complex editing performance on the Level-3 subset of Complex-Edit (Yang et al., 2025b) to assess scalability and effectiveness, as shown in Tab. 3. Our method selects the best-performing tools based on task-specific capability matching, enabling precise execution across diverse editing types. As a result, it achieves the highest scores

Table 1: Basic Image Generation Comparison on T2I-CompBench (Huang et al., 2023)

| Model | Attribute Binding | | | Object Relationship | | Complex ↑ |
|---|---|---|---|---|---|---|
| | Color ↑ | Shape ↑ | Texture ↑ | Spatial ↑ | Non-Spatial ↑ | |
| FLUX (Labs, 2024) | 0.7407 | 0.5718 | 0.6922 | 0.2863 | 0.3127 | 0.3771 |
| SD3 (Esser et al., 2024) | 0.8132 | 0.5885 | 0.7334 | 0.3200 | 0.3140 | 0.3703 |
| GoT (Fang et al., 2025a) | 0.4793 | 0.3668 | 0.4327 | 0.2238 | 0.3053 | 0.3255 |
| T2I-R1 (Jiang et al., 2025) | 0.8130 | 0.5852 | 0.7243 | 0.3378 | 0.3090 | 0.3993 |
| GenArtist (Wang et al., 2024b) | 0.8482 | 0.6948 | 0.7709 | 0.5437 | 0.3346 | 0.4499 |
| T2I-Copilot (Chen et al., 2025a) | 0.8039 | 0.6120 | 0.7604 | 0.3228 | 0.3379 | 0.3985 |
| **Ours (PerfGuard)** | **0.8753** | **0.7366** | **0.8148** | **0.6120** | **0.3754** | **0.5007** |

Table 2: Advanced Image Generation Comparison on OneIG-Bench (Chang et al., 2025)

| Method | Type | Alignment ↑ | Text ↑ | Reasoning ↑ | Style ↑ |
|---|---|---|---|---|---|
| FLUX (Labs, 2024) | Diffusion | 0.786 | 0.523 | 0.253 | 0.368 |
| SD3 (Esser et al., 2024) | Diffusion | 0.801 | 0.648 | 0.279 | 0.361 |
| GoT (Fang et al., 2025a) | CoT | 0.767 | 0.504 | 0.290 | 0.369 |
| T2I-R1 (Jiang et al., 2025) | CoT | 0.793 | 0.662 | 0.297 | 0.370 |
| GenArtist (Wang et al., 2024b) | Agent | 0.747 | 0.501 | 0.285 | 0.352 |
| T2I-Copilot (Chen et al., 2025a) | Agent | 0.821 | 0.679 | 0.318 | 0.386 |
| **Ours (PerfGuard)** | Agent | **0.834** | **0.684** | **0.350** | **0.395** |

Table 3: Complex Image Editing Comparison on Complex-Edit (Yang et al., 2025b)

| Method | IF ↑ | PQ ↑ | IP ↑ | O ↑ |
|---|---|---|---|---|
| AnySD | 4.13 | 7.14 | **9.08** | 6.78 |
| Step1X_Edit | 7.95 | 8.66 | 7.70 | 8.10 |
| GenArtist | 6.14 | 7.24 | 6.19 | 6.52 |
| OmniGen | 7.52 | 8.86 | 8.01 | 8.13 |
| **Ours** | **8.95** | **9.02** | 8.56 | **8.84** |

Table 4: Ablation Study on Design: C., P., and A. stand for CAPO, PASM, and APU.

| C. | P. | A. | Color ↑ | Spatial ↑ | Complex ↑ |
|---|---|---|---|---|---|
| × | × | × | 0.8239 | 0.5600 | 0.4327 |
| ✓ | × | × | 0.8466 | 0.5756 | 0.4493 |
| × | ✓ | × | 0.8521 | 0.5919 | 0.4412 |
| × | ✓ | ✓ | 0.8596 | 0.6005 | 0.4738 |
| **Ours (full)** | | | **0.8753** | **0.6120** | **0.5007** |

in Instruction Following (IF) and Perceptual Quality (PQ). AnySD scores highest in Identity Preservation (IP) due to minimal edits in many Level-3 samples, which also leads to a lower IF score. Overall, our approach outperforms all baselines, demonstrating strong generalization across visual reasoning and generation tasks.

## 5.3 ABLATION STUDY

**Ablation on Design** We performed ablation experiments on the key modules of PerfGuard (Tab. 4), with the results summarized as follows: i) Relying solely on conventional text descriptions for tool capabilities often leads to misselection, forcing the Worker to perform near-exhaustive attempts, resulting in the lowest performance. ii) Even with the Capability-Aligned Planning Optimization mechanism, the lack of an accurate tool selection strategy hinders the Planner's task planning, resulting in limited overall performance improvement. iii) Introducing the Performance-Aware Selection Modeling mechanism significantly improves some metrics, with the color dimension increasing by 3.42% and the texture dimension by 5.7%. iv) Further applying Adaptive Preference Updating fine-tunes preference scores for Planner-generated sub-tasks, enhancing tool selection precision and raising the complex dimension from 0.4412 to 0.4738. v) The performance-driven tool selection strategy improves tool selection accuracy in downstream tasks, enhancing sub-task execution efficiency. This, in turn, boosts overall task planning accuracy by the Planner, further optimizing PerfGuard's performance with CPAO support.

**Capability Matching Method Ablation.** We conducted a systematic evaluation of tool invocation error rates for different capability-matching strategies on the "complex_vel" subset of T2I-CompBench (Fig. 4). The results indicate that relying solely on textual descriptions with QWen3-14B (Yang et al., 2025a) (orange bar) results in a high error rate of 77.8%, due to the presence of similar tools with differing capability focuses, which makes text-based selection unreliable. Even when assisted by the state-of-the-art large language model GPT-4o (Fang et al., 2025b) (yellow bar),

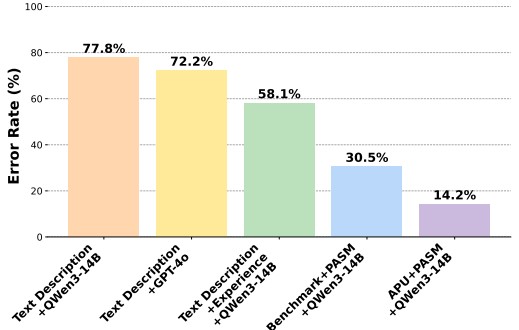

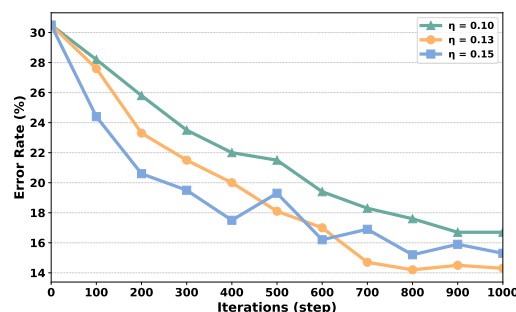

Figure 4: Comparison of capability matching methods. Our method substantially reduces tool selection errors.

Figure 5: Ablation on $\eta$ in Eq. 3. When $\eta = 0.13$, the error rate reaches its minimum of $14.2\%$ at step 800.

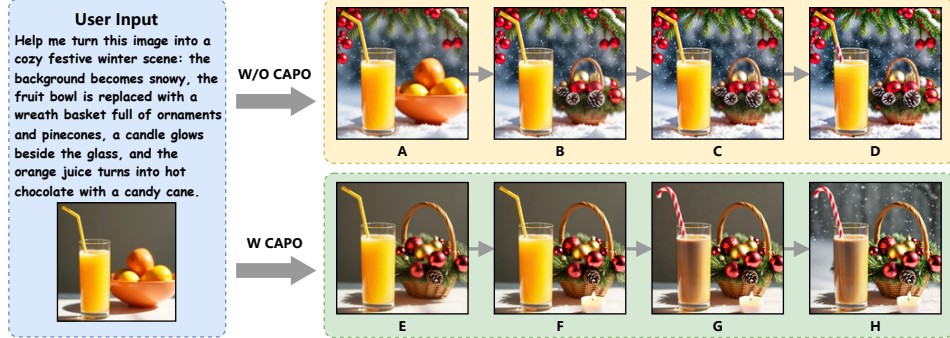

Figure 6: Visualization of the ablation results for CAPO. Operations of Planner: A: Replace the background with a snowy scene. B: Change the fruit bowl to a festive wreath basket with ornaments and pinecones. C: Place a lit candle beside the glass on the table. D: Change the drink to hot chocolate, with a candy cane in place of the straw. E: Swap the fruit bowl for a festive basket of pinecones and decorations. F: Place a candle beside the glass, softly glowing. G: Replace the orange juice with hot chocolate and substitute the straw with a festive candy cane. H: Change the background to a snowy Christmas setting.

the error rate remains high at $72.2\%$, highlighting the limitations of LLMs in interpreting capability descriptions alone. Incorporating an external experience module with QWen3-14B (green bar) reduces the error rate to $68.1\%$ by storing and retrieving historical successful experiences, though the effectiveness is still constrained by retrieval reliability and differences in tool capabilities. Leveraging a benchmark-initialized performance score matrix with QWen3-14B (blue bar) to perform task-specific capability matching significantly lowers the error rate to $30.5\%$. Further applying the Preference Updating mechanism (purple bar) optimizes the error rate to $14.2\%$, demonstrating that capability-aware matching combined with adaptive optimization can effectively enhance the accuracy and robustness of tool selection.

**Ablation on Update Step Size**   To validate the effectiveness of the Adaptive Preference Updating method, as shown in Fig. 5, we studied the impact of different $\eta$ values in Eq. 3 on tool selection error rate using the same dataset as in Fig. 4. Ablation experiments with $\eta$ set to 0.1, 0.13, and 0.15 show that a small $\eta$ (0.1) results in slow error reduction, while a large $\eta$ (0.15) accelerates initial convergence but causes severe oscillations in later stages. In contrast, $\eta = 0.13$ achieves a more efficient and stable decrease, reaching the optimal error rate of $14.2\%$ at step 800. These results indicate that $\eta = 0.13$ provides a balanced trade-off between convergence speed and stability, effectively optimizing tool selection under the current experimental setup.

**Ablation on Capability-Aligned Planning Optimization**   We conducted a visual ablation study on the CAPO to examine the impact of Planner training, as shown in Fig. 6. For fair comparison, we retained only Step1X_Edit in the toolset and removed visual supervision from the Self-Evaluator.

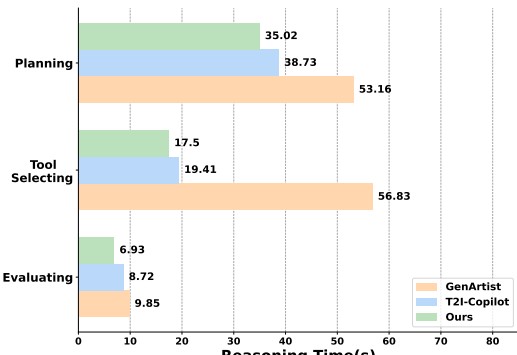

Figure 7: Time consumption of the inference process for different agent methods.

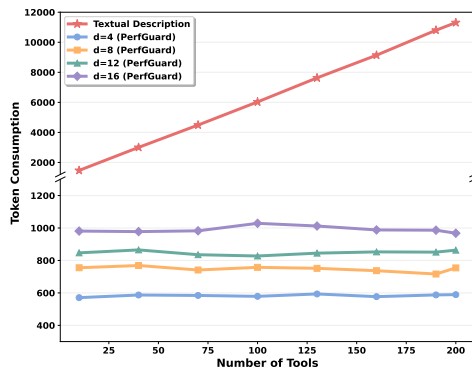

Figure 8: Token consumption comparison of tool selection methods.

Results show that a trained Planner can perceive tool performance boundaries and understand how operation order affects outcomes. For instance, in Fig. 6, editing the background first reduces the success rate of later steps, as Step1X_Edit may introduce inaccuracies that affect other entities like the table. This also suggests that tool limitations can inversely influence planning accuracy.

## 5.4 EFFICIENCY COMPARISON

**Time Consumption Comparison** To validate the inference efficiency of PerfGuard, we uniformly employed QWen3-VL-32B (Qwen3-VL, 2025) as the LLM for PerfGuard, GenArtist, and T2I-Copilot. Inference was performed on the same dataset as Tab. 1, and we recorded the time consumption for task planning, tool selecting, and image evaluatin per round. As shown in Fig. 7, our method exhibits significantly lower time consumption in all three processes compared to counterparts. Particularly, while T2I-Copilot's fixed toolset minimizes its tool selection time, GenArtist's detailed textual descriptions of tool capabilities require more reasoning time when the tool quantity is higher. Conversely, our method, by analyzing sub-tasks and outputting capability-matching preference weights, achieves a tool selection time substantially lower than GenArtist.

**Token Consumption Comparison** To demonstrate the efficiency of our method in tool selection, we expand the problem into large-scale tool management within future agent communities. Specifically, we simulate a large tool library using GPT-4o (Fang et al., 2025b), where the number of tools ranges from 10 to 200, generating tool information with textual descriptions and multi-dimensional ratings. The specific details are provided in the Supplementary Material A.10. We use the "complex_vel" subset from T2I-CompBench for the task and compare PerfGuard's performance-driven tool selection with traditional text-based methods, with a maximum token output of 8192. We compare total token consumption (input and output) between the two methods. Fig. 8 shows: 1) The traditional method consumes more tokens, as it struggles to define tool capabilities, resulting in catastrophic growth in token consumption as the number of tools increases, without addressing selection correctness. 2) PerfGuard, by focusing on task-specific dimensions, is unaffected by the number of tools. 3) As dimensions increase (from d=4 to d=16), token consumption for PerfGuard mainly increases slowly in the input prompts. This demonstrates PerfGuard's superior efficiency in tool management and selection for future agents.

## 6 CONCLUSION

In this work, we address a key challenge in agent-based visual content generation: the lack of precise modeling of tool performance boundaries, which often leads to unreliable planning and inconsistent execution. By incorporating performance-aware mechanisms and feedback-driven refinement, our framework improves decision reliability and strengthens alignment with user-defined goals. These results highlight the importance of bridging tool capability understanding with planning logic. Future efforts will focus on dynamic tool integration and expanding to multimodal tasks to further enhance adaptability and generalization.

ACKNOWLEDGMENTS

This work was supported in part by the Postdoctoral Fellowship Program of the China Postdoctoral Science Foundation (CPSF) under Grant No. GZC20251088.

ETHICS STATEMENT

This work complies with the ICLR Code of Ethics (`https://iclr.cc/public/CodeOfEthics`). Our study does not involve human subjects, sensitive personal data, or any form of biometric information. All datasets used are publicly available and widely adopted in the research community. We have taken care to avoid generating or reinforcing harmful content, stereotypes, or biases in both model design and evaluation. No proprietary or confidential data was used. There are no known conflicts of interest or external sponsorship that could influence the outcomes of this research. We acknowledge the importance of ethical considerations in AI research and have made efforts to ensure transparency, reproducibility, and fairness throughout the development of our framework.

REPRODUCIBILITY STATEMENT

We are committed to ensuring the reproducibility of our work. To facilitate this, we have provided comprehensive implementation details, experimental settings, and evaluation protocols in the main paper and appendix. All datasets used in our experiments are publicly available, and we include detailed data preprocessing steps in the supplementary materials. For our proposed framework and algorithms, we have submitted an anonymous source code repository as part of the supplementary materials, which includes scripts for training, evaluation, and visualization. The repository is available at `https://github.com/FelixChan9527/PerfGuard`. We also provide ablation studies and hyperparameter configurations to support reproducibility. We encourage readers and reviewers to refer to the appendix and supplementary files for further details.

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

# A APPENDIX

## A.1 USE OF LLMS

We use LLMs for research ideation. Details are described in A.2.

## A.2 EXPERIMENTAL SETUP

**Large Language Model Configuration**  PerfGuard employs vLLM (Kwon et al., 2023) as its large language model inference engine and adopts MetaGPT (Hong et al., 2024b) as its underlying framework. For agents responsible for multimodal analysis (Analyst and Evaluator), we use GPT-4o (2024-08-01-preview) (Hurst et al., 2024), whereas agents dedicated to visual reasoning (Planner and Worker) use QWen3-14B (Yang et al., 2025a) for trajectory data collection. The collected trajectories are then used to train QWen3-8B through Capability-Aligned Planning Optimization. During later testing and inference, we replace the Planner's language model with QWen3-8B.

**Tool Library Configuration**  To ensure PerfGuard possesses sufficient visual reasoning capabilities, we configure three types of visual reasoning models in the tool library to validate our approach: "text-to-image tools," "image editing tools," and "customized generation tools." The "text-to-image tools" include FLUX (Labs, 2024), SD3 (Esser et al., 2024), PixArt-$\alpha$ (Chen et al., 2023), and SDXL (Podell et al., 2023); the "image editing tools" include AnySD (Yu et al., 2025), UltraEdit (Zhao et al., 2024), ICEdit (Zhang et al., 2025c), and Step1X_Edit (Liu et al., 2025); the "customized generation tools" include DreamO (Mou et al., 2025), EasyControl (Zhang et al., 2025b), and IPAdapterPlus (Ye et al., 2023).

**Hyperparameter Configuration**  For Adaptive Preference Updating (Eq. 3), we set the number of candidate tools to 3, selecting the top 2 tools by score ($m = 2$) and randomly selecting 1 tool ($n = 1$), with a update step size $\eta = 0.13$. For Capability-Aligned Planning Optimization (Eq. 5), the number of sampled candidate sub-tasks is $k = 5$, and the proportion of experience-based sub-tasks is $\beta = 0.4$.

**Competitors**  i) For the image generation task, we systematically compared three categories of methods: diffusion model-based approaches (e.g., FLUX (Labs, 2024), SD3 (Esser et al., 2024)), Chain-of-Thought (CoT)-based approaches (e.g., GoT (Fang et al., 2025a), T2I-R1 (Jiang et al., 2025)), and agent-based approaches (e.g., GenArtist (Wang et al., 2024b), T2I-Copilot (Chen et al., 2025a)). By contrasting these strategies, we aim to analyze how different visual reasoning mechanisms impact the semantic accuracy of generated images. ii) For the image editing task, we evaluated not only pure diffusion-based methods (e.g., ICEdit (Zhang et al., 2025c), AnySD (Yu et al., 2025)) but also Step1X_Edit (Liu et al., 2025), which integrates large language model (LLM) techniques. To ensure a fair comparison, we additionally included general-purpose models capable of both image generation and editing (e.g., GenArtist and OmniGen (Xiao et al., 2025)).

## A.3 USER STUDY

To validate our method's practical performance, we conducted a user study with 15 non-experts using 20 images from the validation subset of MS-COCO (Lin et al., 2014). Text descriptions were generated by GPT-4o (Fang et al., 2025b), and participants could either input their own text or use these to create matching images. We also tested our method with image-only input. The visualization results are shown in Fig. 9: 1) When using only text, existing methods often lost key details (e.g., foil and "Bubble Up" label in row 2), while our method preserved critical semantics; 2) Interestingly, with image-only input, PerfGuard still achieved accurate generation through fine-grained visual understanding, demonstrating its robust cross-modal understanding capability

We conducted a more detailed evaluation of the images generated by users, which included objective assessments using DINOv2 score (Oquab et al.) (**DINO**, which is used to measure the semantic similarity between the generated image and the reference image.) and CLIP score (Radford et al., 2021) (**CLIP**, which is used to measure the semantic similarity between the generated image and the given text.), as well as subjective evaluations from users. In the subjective evaluation, users rated the generated images on a scale of 1 to 5 based on condition match (**Cond.**, which is the user's score of how well the generated image matches the given conditions.) and aesthetic quality (**Aesthetic**, which is the user's score for the overall aesthetic appeal of the generated image.). After the experiment, users selected the best image generation tool. The experimental results, as shown in Tab. 5, indicate that the objective evaluation results, in terms of image-image consistency and text-image consistency, align closely with the trends presented in the comparative experiment in Tab. 1 and Tab. 2. In the subjective evaluation, GenArtist scored the lowest in condition match and aesthetic appeal due to its lack of accurate understanding and optimization of information. T2I-Copilot, which focuses on image generation tasks, performed better in condition match and aesthetic appeal by optimizing and

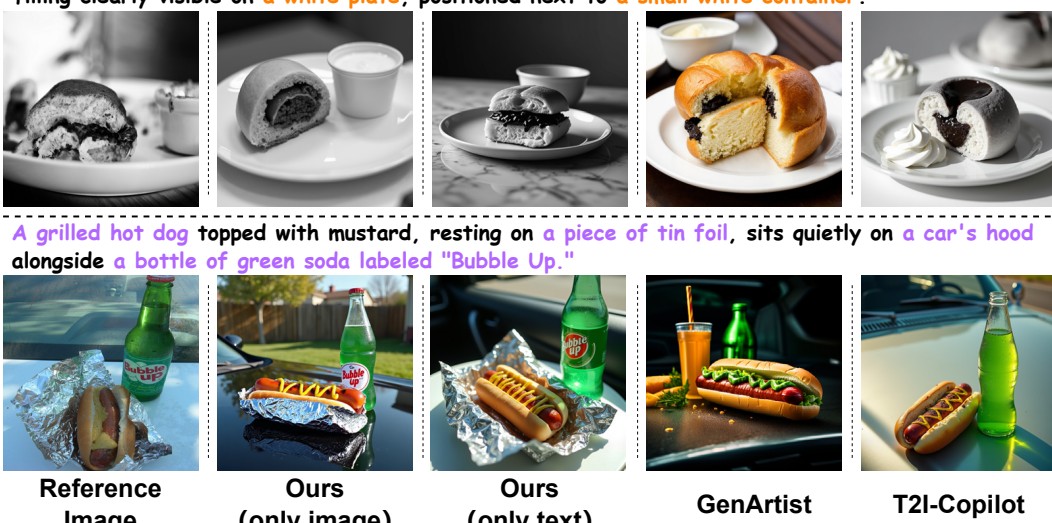

Figure 9: **The visualization of the user study, where PerfGuard, GenArtist, and T2I-Copilot are used to generate output images that are most similar to the reference images from the MS-COCO dataset. Among them, PerfGuard compares the results generated using only image input and only text input.**

Table 5: User study of different methods.

| Method | DINO | CLIP | Condition | Aesthetic | User-Pref |
|---|---|---|---|---|---|
| GenArtist | 0.7440 | 0.3401 | 3.15 | 2.53 | 6.7% |
| T2I-Copilot | 0.8134 | 0.3652 | 3.42 | 3.69 | 20.0% |
| Ours (Only text) | 0.8467 | 0.3723 | 3.67 | 3.88 | **73.3%** |
| **Ours (Only image)** | **0.8716** | **0.3962** | **3.80** | **4.12** | |

enriching the input information. However, these methods lack accurate understanding of the information and tool selection during the actual generation process, which is why PerfGuard achieved the best subjective and objective results. Furthermore, we asked users to choose the best tool for each image generated by the three methods and recorded the proportion of users who favored each tool (**User-Pref**). Among the tested samples, 73.3% of users chose PerfGuard, indicating that our method provided the best user experience across various input formats.

## A.4 PERFORMANCE COMPARISON OF AGENTS USING THE SAME LLM

To balance multimodal analysis and inference performance while ensuring fairness in performance comparison, we replaced the LLMs of GenArtist, T2I-Copilot, and PerfGuard with Qwen3-VL-32B and compared their performance on image generation tasks. The remaining experimental configurations and datasets were consistent with those in Tab. 1. The experimental results, shown in Tab. 6, indicate the following: 1) GenArtist and T2I-Copilot heavily rely on the performance of closed-source LLMs for task analysis and planning. As a result, due to their lower LLM generalization ability, their performance metrics suffer a significant decline. 2) In our approach, the closed-source MLLM is only used as an image interpreter to assist the LLM in performing the analysis and planning process. Therefore, when all LLM modules are replaced with Qwen3-VL-32B, the performance degradation is minimal. 3) Even when GPT-4o is replaced with Qwen3-VL-32B, the trend remains consistent with that shown in Tab. 1, where our method outperforms GenArtist and T2I-Copilot across multiple metrics.

Table 6: Performance comparison of agents using the same LLM.

| Method | Color | Spatial | Complex |
|---|---|---|---|
| GenArtist (Qwen3-VL-32B) | 0.5670 | 0.2928 | 0.2321 |
| T2I-Copilot (Qwen3-VL-32B) | 0.6755 | 0.2257 | 0.2461 |
| Ours (Qwen3-VL-32B) | 0.8500 | 0.5481 | 0.4538 |
| **Ours (Original Config)** | **0.8753** | **0.6120** | **0.5007** |

Table 7: Performance comparison of PerfGuard using different LLMs

| Method | Color | Spatial | Complex | Degradation (Complex) – |
|---|---|---|---|---|
| GenArtist (Qwen3-VL-32B) | 0.5670 | 0.2928 | 0.2321 | 48.4% |
| GenArtist (Original Config) | 0.8482 | 0.5437 | 0.4499 | – |
| T2I-Copilot (Qwen3-VL-32B) | 0.6755 | 0.2257 | 0.2461 | 38.2% |
| T2I-Copilot (Original Config) | 0.8039 | 0.3228 | 0.3985 | – |
| Ours (Qwen3-VL-32B) | 0.8500 | 0.5481 | 0.4538 | **9.4%** |
| Ours (GPT-4o) | 0.8577 | 0.6004 | 0.4813 | **3.9%** |
| **Ours (Original Config)** | **0.8753** | **0.6120** | **0.5007** | – |

## A.5 ABLATION STUDY WITH DIFFERENT LLMs

We validated the generalization and robustness of the PerfGuard framework across different LLMs by replacing the MLLM configuration with either the closed-source GPT-4o or the open-source Qwen3-VL-38B. The ablation study followed the same experimental setup as the comparison experiments in Tab.1, with the only difference being the replacement of the MLLM. Since GPT-4o is a closed-source model, we could not implement the CAPO training process as originally proposed. Therefore, an experience replay mechanism was adopted to ensure that the performance of Perf-Guard based on GPT-4o closely matched the planning accuracy of CAPO. The experimental results, shown in Tab. 7, indicate that replacing all PerfGuard modules with Qwen3-VL-38B or GPT-4o led to a slight performance degradation. Compared to the performance degradation rates of GenArtist and T2I-Copliot, our method shows better adaptability to different LLMs. For instance, when using Qwen3-VL-38B in the Complex dimension of T2I-CompBench, GenArtist's performance degrades by about 48.4%, T2I-Copliot's by about 38.2%, while PerfGuard's performance degrades by only 9.4%. This demonstrates that the proposed method achieves superior performance across various LLM settings (both open-source and closed-source), further confirming its robustness and generalization capability across different LLMs.

## A.6 LIMITATIONS AND CHALLENGES

The PerfGuard method we proposed is a preliminary attempt to address the issue of accurate tool selection in agent systems. However, there are still limitations and challenges that need to be solved: the Performance-Aware Selection Modeling strategy we proposed relies on existing benchmark scores to initialize the tool performance boundary matrix. However, in domains outside of visual content generation, high-quality benchmarks may not always be available for efficient initialization of the capability boundary matrix. Therefore, future work will focus on expanding this method to other domains, such as visual reasoning tasks, and advancing it to the level of multi-agent capability discovery for better multi-agent and multi-tool capability matching and collaboration.

## A.7 DESIGN AND SPECIFICATION OF PERFORMANCE BOUNDARIES IN PERFGUAR

We initialize tool performance boundaries by leveraging existing multi-dimensional evaluation benchmarks conducted on large-scale datasets. Specifically, we adopt the evaluation dimensions from t2i-compbench (Huang et al., 2023) and ImgEdit-Bench (Ye et al., 2025) as the performance boundary dimensions for image generation and editing tools in the library, respectively. The detailed definitions are as follows:

**Image generation performance boundary dimensions:**

- **color**: indicates the accuracy of the object's color in the generated image.
- **shape**: indicates the accuracy of the object's shape in the generated image.
- **texture**: indicates the accuracy of the object's material or surface quality in the generated image, such as "wooden", "metallic", etc.
- **2D-spatial**: indicates the accuracy of the 2D spatial relationships between objects in the generated image, such as "on the side of", "on the left", "on the top of", "next to", etc.
- **3D-spatial**: indicates the accuracy of the 3D spatial relationships between objects in the generated image, such as "behind", "hidden by", "in front of", etc.
- **numeracy**: indicates the accuracy of the number of objects in the generated image.
- **non-spatial**: indicates the accuracy of non-spatial relationships between objects in the generated image, such as "A is holding B", "C is looking at D", "E is sitting on F", etc.

**Image editing performance boundary dimensions:**

- **addition**: indicates the accuracy of adding objects to the image.
- **removement**: indicates the accuracy of removing objects from the image.
- **replacement**: indicates the accuracy of replacing objects in the image.
- **attribute-alter**: indicates the accuracy of modifying the attributes of objects in the image.
- **motion-change**: indicates the accuracy of modifying the actions, movements, or spatial positions of objects in the image.
- **style-transfer**: indicates the accuracy of modifying the overall style of the image.
- **background-change**: indicates the accuracy of modifying the background of the image.

## A.8 DEFINITION OF $\mathcal{R}_{\text{ACTUAL}}$ IN ADAPTIVE PREFERENCE UPDATING

For the actual usage ranking $\mathcal{R}_{\text{actual}}$ in Eq.3, we employ GPT-4o to directly compare the outputs of multiple candidate tools and evaluate their effectiveness in executing the given subtask. The prompt is configured as follows:

| Prompt Engineering of Actual Usage Ranking |
| --- |
| task: task
Multiple output images are generated after executing the task. Please compare ONLY these output images and analyze to provide their ranking from best to worst.
Do not include any images outside of this list in your analysis or ranking.
If multiple images meet the task criteria equally well, prioritize the image that appears most natural and visually coherent. |

## A.9 PROMPT ENGINEERING

The agent system designed in this work consists of four roles: Analyst, Planner, Worker, and Self-Evaluator. Their prompt engineering strategies will be presented in this subsection.

| Prompt Engineering of Analyst |
| --- |
| 1. Please analyze the user's needs based on the provided content and summarize their requirements.
If a specific image is referenced, the path to the reference image must be specified.
**No assumptions are allowed about the user-provided information; the output must closely align with the user's given information.**
**The output must be derived through precise and correct reasoning, rather than copying the user's input.**
Transform the user input into concrete visual elements for the final image, avoiding overly simple or abstract terms. |

The output task must be **precise and concise, within 20 tokens**.
Output the task in the format: <task>Your summary task </task>.
2. Please provide the semantics of the final output image (i.e., what the final rendered image looks like) in textual form.
The output semantic should be described in terms of key objects in the image, their attributes (numeracy, categories, color, texture, etc.), spatial relationships, background, and image style, etc..
The output semantic must be **precise and concise, within 20 tokens**.
Output the semantic in the format: <semantic>Textual semantic information of the target image. </semantic>.

---

**Prompt Engineering of Planner**

Task:
Current image semantics:
Target image semantics:
==================================================
Available features:
1. Image generation: Create an image strictly matching the target semantics. Specify only required dimensions: quantity (use "exactly" if needed), position, attributes, material, color, style, lighting, or semantic relationships.
2. Image editing: Modify an existing image to gradually match target semantics. Adjust only necessary dimensions; do not add unrelated objects. Regeneration of the whole image is not allowed.
Instructions:
- Analyze the **target image semantics**, **task requirements**, and **historical operation information** (if available).
- **Provide the next processing step to gradually meet the final task requirements through subsequent multi-round interactions.**
- Each operation should be concise (less 30 words) while retaining essential elements.
- Use precise instructions (e.g., "remove the apple on the far right"), avoiding vague expressions.
- Preferably output a single most effective operation per round; if the task is complex and model capability allows, multiple operations can be included in one round.
- For generation tasks, output images should be natural and harmonious.
- For editing tasks, do not regenerate images arbitrarily; only modify necessary parts.
- If multiple operations can achieve the task, select the one with the highest success rate.
- Specify dependencies clearly: `<depend>None</depend>` if independent, or `<depend>round X</depend>` if based on a previous round.

---

**Prompt Engineering of Worker**

Task:
1. You have two types of tools to choose from: **text2image-generation tool**, **image-editing tool**. Please choose the appropriate tool-type based on the task requirements. Output in XML format:
`<category>your select tools-type</category>`
2. If you choose the `text2image-generation tool` category, analyze the task and assign weights for the following preferences:
`'color', 'shape', 'texture', '2D-spatial', '3D-spatial', 'numeracy', 'non-spatial'`

- `color`: object's color requirement
- `shape`: object's shape requirement
- `texture`: material/surface quality (e.g., wooden, metallic)
- `2D-spatial`: 2D spatial relationships (e.g., on the left, next to)

- `3D-spatial`: 3D spatial relationships (e.g., behind, in front of)
- `numeracy`: number of objects
- `non-spatial`: non-spatial relationships (e.g., A is holding B)

3. If you choose the `image-editing tool` category, analyze the task and assign weights for:
`'addition', 'removement', 'replacement', 'attribute-alter', 'motion-change', 'style-transfer', 'background-change'`

- `addition`: adding objects
- `removement`: removing objects
- `replacement`: replacing objects
- `attribute-alter`: modifying object attributes
- `motion-change`: changing actions or positions
- `style-transfer`: modifying image style
- `background-change`: modifying background

**Notes:**

- Weights range from 0 to 1; higher values indicate greater importance.
- The sum of all weights must be 1.
- Assign very low values (even 0) to unimportant dimensions.
- Ensure weights strictly reflect task requirements.
- Do not confuse preferences between the two tool types.

A.10 CONFIGURATIONS RELATED TO THE TOOL SELECTION SIMULATION EXPERIMENT IN SEC. 5.4

The tool selection prompt engineering based on textual descriptions.

> The tool selection prompt engineering based on textual descriptions.
>
> You are an expert tool selection agent.
> Task Description:
> {task_description}
> Below is the complete list of available tools in your tool library.
> These tools are provided in the attached file "{tool_file}".
> ————————— TOOLS BEGIN —————————
> {tools_text}
> ————————— TOOLS END ——————————-
> Each tool includes:
> - A precise tool name
> - A description of its capabilities
> - The type of task it is designed for
> Your objective:
> Carefully analyze the task and select the SINGLE most appropriate tool.
> Instructions for reasoning:
> 1. For each tool in the library, analyze whether it is suitable for the given task.
> 2. For each tool, provide a brief reasoning:
> - Why this tool is suitable (or not suitable) for the task.
> 3. After analyzing all tools, give your final choice of the SINGLE most appropriate tool.
> 4. Output the final answer ONLY as an XML tag in the following form:
> `<tool>YOUR_CHOSEN_TOOL_NAME</tool>`
> Rules:

- You must reason about all tools individually before making the final choice.
- Provide clear and concise reasoning for each tool.
- Do not skip any tool in the analysis.
- The XML output at the end must exactly match the chosen tool's name.
- No explanation outside of the tool analysis and final XML output.

The tool selection prompt engineering based on performance-driven selection.

> ### The tool selection prompt engineering based on performance-driven selection.
>
> You are an expert tool selection agent.
> Task Description:
> {task_description}
> Your objective:
> 1. Carefully analyze the task requirements.
> 2. Analyze the overall task, firstly.
> 3. For ALL relevant dimensions (both Text2Image and ImageEditing dimensions):
> - Analyze the importance of this dimension for the given task.
> - Assign a weight between 0 and 1 based on its importance.
> - Provide a brief reasoning for the assigned weight.
> Text2Image dimensions: {TEXT2IMAGE_DIMENSIONS}
> ImageEditing dimensions: {IMAGE_EDIT_DIMENSIONS}
> Instructions for output:
> 1. First, reason about each dimension individually. For example:
> Dimension: color
> Importance: High
> Reason: The task emphasizes vivid and harmonious colors in the wizard's clothing and magical effects.
> Assigned weight: 0.25
> 2. Repeat for all dimensions, even if the weight is 0.
> 3. After analyzing all dimensions, give the final category and weights in **exact XML format**:
>
> ```
> <category>TOOL_CATEGORY</category>
> <preference>
> <dim1>weight</dim1> <!-- Reason: explanation for weight -->
> <dim2>weight</dim2> <!-- Reason: explanation for weight -->
> ...
> </preference>
> ```
>
> Rules:
> - Weights must sum to 1.
> - Use all relevant dimensions for weighting; irrelevant dimensions can have weight 0.
> - Provide concise reasoning for each dimension in the XML comment.
> - Only output the XML; do not include explanations outside the XML.

## A.11   MORE VISUALIZATION RESULTS

We conducted extensive visualizations of PerfGuard across various image-related tasks. Fig. 10 presents multiple examples from the image editing task, Fig. 11 showcases several cases from the text-to-image generation task, Fig. 12 illustrates a range of customized image generation results, and Fig. 13 depicts multiple instances of PerfGuard's error correction workflow during image generation.

Modify the original image by switching the wooden platter to one with a polished marble finish, removing the middle fork, and changing the chocolate cupcake to a lighter caramel color.

| Original image | Replace the platter with polished marble. | Remove the middle fork. | Lighten the cupcake to caramel. |

Edit the original image by changing the clothing to a denim jacket texture, replacing the background with a concert stage, and updating the guitar body to a vibrant green.

| Original image | Change clothing to denim texture. | Replace background with concert stage. | Update guitar color to vibrant green. |

Modify the original image to make the fire hydrant appear aged with rust, change the woman's shirt to a soft-blue color, and set the scene against a cityscape background.

| Original image | Age the fire hydrant with rust. | Change the woman's shirt to soft-blue. | Replace the background with a cityscape. |

Edit the original image by changing the chocolate cake to blue, removing the background flags, and adding a flower bouquet with falling confetti.

| Original image | Recolor the cake to blue. | Remove the background flags. | Add a bouquet and falling confetti. |

Edit the original image by redesigning the fire hydrant with a star pattern, replacing the ground with lush green grass where a white rabbit sits, and setting the scene at dusk.

| Original image | Redesign the fire hydrant with a star pattern. | Replace the ground with green grass and add a white rabbit. | Change the time of day to dusk. |

Figure 10: Visualization examples of the image editing task

Create an image of a woman in a yellow hat and dress sitting on a garden bench, holding a basket of red roses. The scene should have a classic, romantic ambiance.

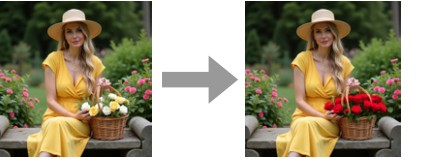 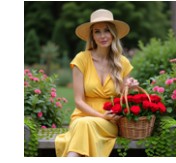 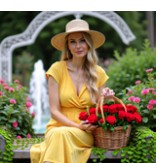

A woman in a yellow hat and dress sits on a stone bench in a garden, holding a basket of roses.

Change the roses in the basket to red roses.

Cover the bench with ivy.

Add a white wrought-iron arch and a fountain to the background.

Create a detailed close-up image of a rustic, weathered wooden wall that showcases its grain and knots. The image should convey a sense of authenticity and being well-used, with faint water stains along the top indicating its age and the effects of the elements.

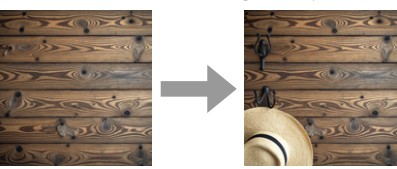 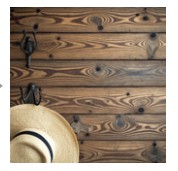 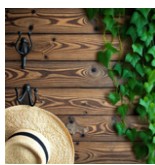 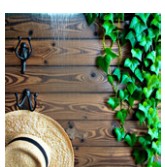

A rustic wooden wall with weathered planks, visible wood grain, knots, and warm brown tones.

Mount two vintage metal hooks on the left and hang a woven straw hat below them.

Add trailing ivy growing from the right-side gap of the wall.

Include subtle water stain marks along the top edge of the wall.

Generate a serene landscape painting. The scene should feature a winding stone path leading through foreground flowers towards cozy red-roofed houses nestled among trees, with hazy mountains in the distance. An elderly person should be seen working in a field nearby.

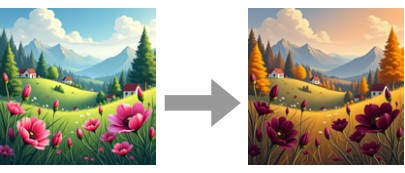 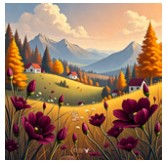 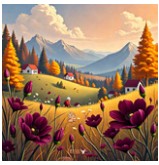

A serene landscape with vibrant magenta flowers, trees, nestled red-roofed houses, distant hazy mountains, and a blue sky.

Change season to autumn.

Add a winding stone path.

An elderly person is working in a field near the houses.

Generate a tranquil, atmospheric night scene of a modern bedroom. The composition should focus on a nightstand by a large window, holding a deep blue backpack, a soft pink toothbrush, and a glowing smartphone. Outside the window, show a rainy night view.

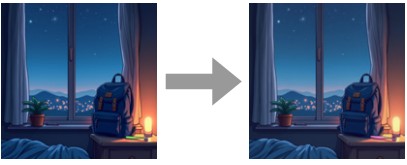 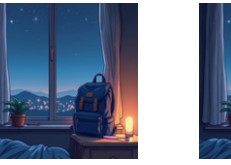 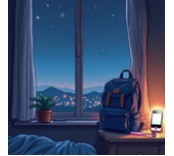 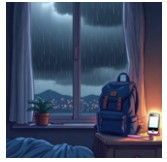

A tranquil night scene of a room with a deep blue backpack and a toothbrush on a nightstand, by a large window.

Change the toothbrush color to a soft pink.

Add a glowing smartphone on the nightstand.

Replace the starry sky with a rainy window view.

Figure 11: Visualization examples of the text to image generation task

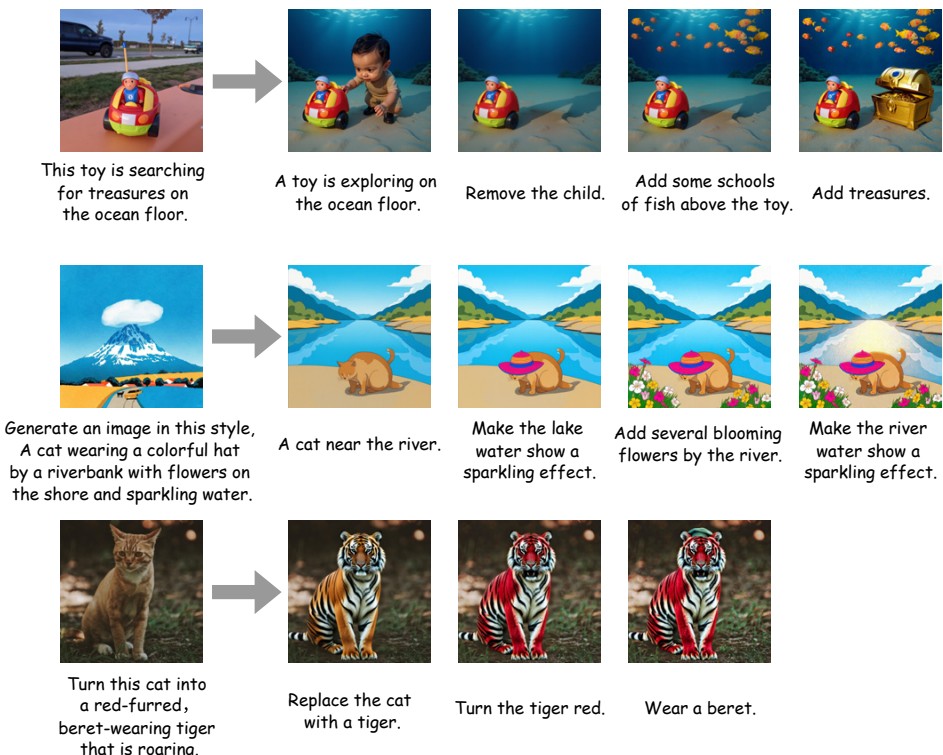

Figure 12: Visualization examples of the customized image generation task

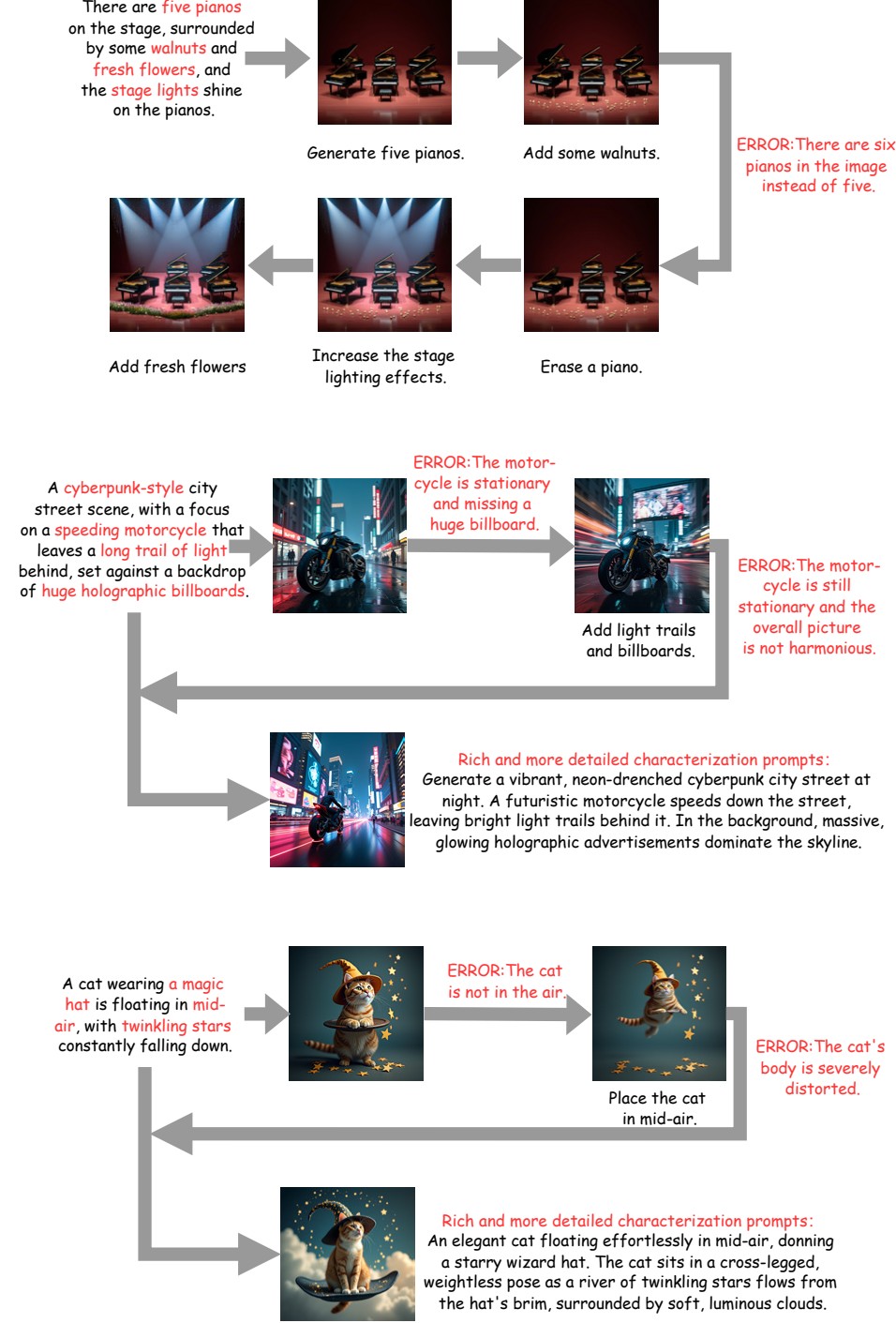

Figure 13: Visualization examples of multiple instances of PerfGuard's error correction workflow during image generation

