# OpenReview forum: "PerfGuard: A Performance-Aware Agent for Visual Content Generation"
_ICLR.cc/2026/Conference — ICLR 2026 Poster_

### Official Review · Reviewer_SDJK · 2025-10-30

**Soundness:** 3
**Presentation:** 2
**Contribution:** 2
**Rating:** 6
**Confidence:** 2

**Summary:**

This paper presents PerfGuard, a performance-aware agent framework for visual content generation. The key insight is that existing agent systems assume tools always work perfectly and use vague text descriptions that don't specify what each tool actually does well. To fix this, PerfGuard introduces three mechanisms: (1) PASM replaces text descriptions with multi-dimensional performance scores, (2) APU dynamically adjusts these scores based on actual execution results, and (3) CAPO aligns the planning process with performance-aware tool selection. Experiments on three benchmarks show improvements in tool selection accuracy and output quality for both image generation and editing tasks.

**Strengths:**

- The paper tackles an overlooked problem in agent-based visual generation. Most prior work assumes tools execute reliably, but this paper explicitly models tool performance boundaries. The combination of PASM, APU, and CAPO feels well thought out and addresses the problem systematically rather than with isolated fixes.

- The experiments are thorough. The authors compare against multiple baselines including diffusion models, CoT-based methods, and other agent systems across different task types. The ablation studies clearly show what each component contributes. I also appreciate the hyperparameter analysis, which helps with reproducibility.

- The paper is well organized. The framework architecture is clearly explained, with distinct agent roles and mechanisms. Figures 1 and 2 effectively illustrate the workflow. The appendix provides good implementation details.

**Weaknesses:**

1. The tool library only includes a specific set of popular models like FLUX, SD3, and Step1X Edit. I'm curious why certain relevant tools are excluded. For example, what about newer 2025 models or domain-specific editing tools? This raises questions about how well the framework generalizes to different tool ecosystems.

2. When adding new tools, the method initializes performance scores by averaging similar tools. This seems reasonable but might miss unique capabilities of novel tools. Have the authors tried other initialization strategies, like few-shot evaluation or transfer learning? A comparison would strengthen this design choice.

3. The quantitative metrics are comprehensive, but there's no user study. Since visual generation is inherently subjective, getting human feedback on the outputs would make the claims more convincing. Even a small preference study would help.

4. How does PerfGuard scale when the tool library grows large? If you have hundreds of tools, does the multi-dimensional scoring and adaptive updating become computationally expensive? The paper doesn't really discuss this.

**Questions:**

1. What criteria did you use to select tools for the library? Were any potentially useful tools excluded, and if so, why?

2. You found η=0.13 works best in your experiments. Does this parameter need tuning for different task types or tool libraries? How would users know what value to use?

3. Any plans to add user studies? It would be interesting to see if humans prefer PerfGuard's outputs over baselines.

4. How does computational cost scale with tool library size? What's the latency for tool selection and planning compared to baseline agent systems?

5. Could the performance dimensions extend to other visual tasks like video generation or 3D content creation? What modifications would that require? (Just for discussion)

---

> ### Author Response · Authors · 2025-11-20
> **Response to Reviewer SDJK (part1)**
>
> **Weaknesses-Question 1: The standards for tool selection in PerfGuard. Are there any tools that are excluded? Additionally, what is the generalization capability of the framework for different tool ecosystems?**
>
> **Answer:** We appreciate the reviewer's interest in the tool selection criteria and the generalization capability of PerfGuard. We address these points as follows:
>
> 1. **Tool Selection Criteria**: In our experiments, we prioritized text-based image generation and editing tools to avoid fairness issues. For example, some image editing tools support multiple input modalities, but using multi-modal inputs with text-based image editing tools is often more challenging. This makes it difficult to distinguish whether poor performance in the generated image is due to the tool's inherent limitations or the complexity of its usage. By focusing on text-based tools, we ensure that performance issues are more clearly attributed to tool capabilities rather than input modality complexity.
>
> 2. **Adaptability to Tools**: PerfGuard is adaptable to any tool. Our tool library includes text-based image generation tools, text-instruction-based editing tools, and customized image generation tools, totaling 12 diverse tools, as shown in the table below. Although our library may lack tools for other input modalities, this does not affect the results or conclusions.
>
>    | Text to Image Genernation Tools | Instruction-guided Image Editing Tools | Customized Image Generation Tools |
>    | ------------------------------- | -------------------------------------- | --------------------------------- |
>    | FLUX                            | AnySD                                  | DreamO                            |
>    | SD3                             | ICEdit                                 | EasyControl                       |
>    | PixArt-α                        | Step1X_Edit                            | IPAdapterPlus                     |
>    | SDXL                            | UltraEdit                              | Emu2                              |
>
>
>
> 3. **Generalization Across Tool Ecosystems**: PerfGuard supports adding new tools, including those with non-text inputs, by adjusting the reasoning process and incorporating extra tools as needed. For example, the CreatLayout tool was added to assist with accurate location generation. Despite its initial design for bounding boxes as input, prompt engineering enabled smooth integration for the task-specific use case, demonstrating the framework’s flexibility across different tool ecosystems.
>
> ---
>
> **Weaknesses 2: The addition of new tools with the performance boundary matrix initialized using average values may overlook the unique features of the new tools.**
>
> **Anwser:** Regarding the initialization of the performance boundary matrix for newly added tools, we respond as follows:
>
> 1. For newly added tools, since they have not undergone detailed performance evaluation, their specific tool characteristics are unknown. Therefore, using average initialization ensures that the new tool's performance is set at the proposed average level, preventing it from being overlooked during the tool selection process.
>
> 2. Once the performance boundary matrix is constructed based on actual tasks, for newly added tools, even if initialized with average values, the tools typically converge after about 150–200 calls during the APU (score update) phase. This process is executed during actual task execution, allowing for continuous learning through an execute-evaluate-update loop, which keeps the training cost low.
>
> 3. We agree with the reviewer’s suggestion that methods such as few-shot evaluation and transfer learning could be beneficial for the initialization of the performance boundary matrix. However, our current focus in this paper has not been on the initialization of the performance boundary matrix. In future research, we will explore this issue in more depth and consider incorporating few-shot evaluation or transfer learning techniques to further enhance the learning efficiency of the boundary matrix.

---

> ### Author Response · Authors · 2025-11-20
> **Response to Reviewer SDJK (part2)**
>
> **Question 2: The determination of the optimal value for the hyperparameter $\eta$.**
>
> **Anwser:** We will address this issue as follows:
>
> 1. The value of $\eta = 0.13$ in this paper was determined through experimental trials. $\eta$ represents the iteration step size for the tool performance matrix scores. Our ablation studies show that a larger $\eta$ (e.g., 0.15) tends to cause oscillations in the score iteration, while a smaller $\eta$ results in slower convergence. To demonstrate optimal performance, we used the $\eta$ value of 0.13, which balances both convergence speed and score quality.
> 2. We acknowledge that the determination of the $\eta$ value may depend on the specific domain involved. However, in the process of updating the performance boundary matrix, the matrix values are normalized to a range of 0 to 1. Therefore, in practical applications, smaller values of $\eta$​ can be set to allow for more iterations, helping the performance boundary matrix to achieve better convergence.
>
> ---
>
> **Weaknesses-Question 3:  A user study is required to explore user preferences for different methods, further enhancing the persuasiveness of the conclusions from a subjective perspective.**
>
> **Anwser:** Good idea! To further analyze the preferences and actual execution performance of our proposed PerfGuard compared to other methods for visual content generation tasks, we conducted a user study with 15 non-experts using 20 images from the validation subset of MS-COCO. Text descriptions were generated by GPT-4o, and participants could either input their own text or use these to create matching images. We also tested our method with image-only input. We conducted a detailed evaluation of the images generated by users, which included objective assessments using DINOv2 score (**DINO**, which is used to measure the semantic similarity between the generated image and the reference image.) and CLIP score (**CLIP** which is used to measure the semantic similarity between the generated image and the given text.), as well as subjective evaluations from users. In the subjective evaluation, users rated the generated images on a scale of 1 to 5 based on condition match (**Cond.**, which is the user's score of how well the generated image matches the given conditions.) and aesthetic quality (**Aesthetic**, which is the user's score for the overall aesthetic appeal of the generated image.). After the experiment, users selected the best image generation tool. The experimental results are as follows:
>
> | Method            | DINO       | CLIP       | Condition | Aesthetic | User-Pref |
> | ----------------- | ---------- | ---------- | --------- | --------- | --------- |
> | GenArtist         | 0.7440     | 0.3401     | 3.15      | 2.53      | 6.7%      |
> | T2I-Copilot       | 0.8134     | 0.3652     | 3.42      | 3.69      | 20.0%     |
> | Ours (Only text)  | 0.8467     | 0.3723     | 3.67      | 3.88      | **73.3%** |
> | Ours (Only image) | **0.8716** | **0.3962** | **3.80**  | **4.12**  | **73.3%** |
>
> The experimental results indicate that the objective evaluation results, in terms of image-image consistency and text-image consistency, closely align with the trends presented in the comparative experiments (originally in Tab. 1 and Tab. 2). In the subjective evaluation, GenArtist scored the lowest in condition match and aesthetic appeal due to its lack of accurate understanding and optimization of information. T2I-Copilot, which focuses on image generation tasks, performed better in condition match and aesthetic appeal by optimizing and enriching the input information. However, these methods lack accurate understanding of the information and tool selection during the actual generation process, which is why PerfGuard achieved the best subjective and objective results. Furthermore, we asked users to choose the best tool for each image generated by the three methods and recorded the proportion of users who favored each tool (**User-Pref**). Among the tested samples, 73.3\% of users chose PerfGuard, indicating that our method provided the best user experience across various input formats. In the supplementary material **A.3** of the revised paper, we provide a detailed explanation of the user study, along with additional visual comparisons.

---

> ### Author Response · Authors · 2025-11-20
> **Response to Reviewer SDJK (part3)**
>
> **Weaknesses-Question 4:  When the tool library contains a large number of tools (hundreds), how does PerfGuard's scalability and computational cost compare to the baseline agent's inference time?**
>
> **Answer:** Thank you for your insightful and valuable question! We address your concerns as follows:
>
> 1. **Cost of Tool Score Updates:** In the iterative process of updating the performance boundary matrix scores (APU), the computational cost per round can be controlled by the number of tools sampled by the user. A larger number of sampled tools results in a higher computational cost per update step, but also leads to more accurate score updates for the performance boundary matrix.
> 2. **Managing New Tools:** In practice, new tools typically perform better than older ones. Therefore, the continuous expansion and updating of the tool library can be managed more flexibly by introducing a selection mechanism. This can include discarding outdated or low-performance tools, or adjusting the selection probability of older tools during the score iteration process. Such strategies help reduce the computational cost of score updates.
> 3. **Computational Cost with Tool Library Scale:** To understand how computational costs scale with the size of the tool library, we designed a set of simulation experiments. These experiments explore the computational cost of both traditional text-based tool capability description methods and our performance-driven tool selection approach. Specifically, we simulated a large tool library using GPT-4o, with the number of tools ranging from 10 to 200, generating tool information with textual descriptions and multi-dimensional ratings. The details of the experiment are provided in Supplementary Material A9. We used the "complex_vel" subset from T2I-CompBench for the task and compared the performance-driven tool selection of PerfGuard with traditional text-based methods. The comparison focused on total token consumption (both input and output), with a maximum token output of 8192. The experimental results are as follows:
>
> | Method/Tool Num | 10 | 40 | 70 | 100 | 130 | 160 | 190 | 200 |
> | :--- | :--- | :--- | :--- | :--- | :--- | :--- | :--- | :--- |
> | Textual Description Method | 1469.3 | 3002.9 | 4489.6 | 6030.3 | 7630.3 | 9139.3 | 10802.6 | 11308.7 |
> | Ours (d=4) | 570.8 | 587.0 | 584.3 | 578.8 | 593.5 | 576.9 | 587.6 | 589.3 |
> | Ours (d=8) | 755.0 | 768.8 | 741.2 | 757.6 | 751.5 | 736.8 | 716.4 | 754.1 |
> | Ours (d=12) | 846.9 | 865.0 | 835.4 | 827.5 | 845.0 | 852.8 | 851.3 | 863.5 |
> | Ours (d=16) | 981.1 | 977.8 | 982.6 | 1028.9 | 1012.3 | 988.1 | 986.5 | 967.9 |
>
> As shown in the table: 1) The Text-based tool selection method consumes more tokens, as it struggles to define tool capabilities, leading to a rapid increase in token consumption as the tool library grows, without considering selection accuracy. 2) PerfGuard, by focusing on task-specific dimensions, remains unaffected by the number of tools. 3) As the dimensions increase from d=4 to d=16, token consumption for PerfGuard mainly increases slowly in the input prompts. This demonstrates PerfGuard's superior efficiency in tool management and selection. In the revised version of the paper, we provide a more intuitive and detailed description in **Sec. 5.4**, "Experiment Efficiency Comparison," with visualizations.
>
> 4. **Inference Efficiency Comparison**: To validate the inference efficiency of PerfGuard, we used QWen3-VL-32B as the LLM for PerfGuard, GenArtist, and we recorded the time consumption for task planning, tool selection, and image evaluation for each round. The inference time distribution for each part of the agent is compared as follows:
>
> | Method      | Planning (s)  | Tool Selecting (s) | Evaluating (s) |
> | ----------- | --------- | -------------- | ---------- |
> | t2i-copilot | 38.73     | 19.41          | 8.72       |
> | genartist   | 53.16     | 56.83          | 9.85       |
> | **Ours**        | **35.02** | **17.50**      | **6.93**   |
>
> As shown in table, our method exhibits significantly lower time consumption in all three processes compared to counterparts. Particularly, while T2I-Copilot's fixed toolset minimizes its tool selection time, GenArtist's detailed textual descriptions of tool capabilities require more reasoning time when the tool quantity is higher. Conversely, our method, by analyzing sub-tasks and outputting capability-matching preference weights, achieves a tool selection time substantially lower than GenArtist.  The above content is explained in detail in **Sec. 5.4** "Efficiency Comparison" of the revised version of the paper.

---

> ### Author Response · Authors · 2025-11-20
> **Response to Reviewer SDJK (part4)**
>
> **Question 5: Is PerfGuard scalable to other visual tasks, and what modifications would be necessary?**
>
> **Anwser:** Regarding the question of whether PerfGuard can be scaled to other visual tasks, our answer is affirmative. When PerfGuard needs to be extended to other visual tasks, the following process can be followed. We will explain this using the 3D generation domain as an example.
>
> 1. Since the context engineering in this paper is primarily designed for image generation and editing tasks, when extending to video generation or 3D generation domains, one can refer to the operations in references [1] and [2], dividing the generation task into an image semantic reasoning process and a modeling process.
>
> 2. Once the user has organized the tool invocation code according to our tool template requirements, only minor modifications are needed in the planner's prompt, such as adding "first perform static image reasoning, then execute image modeling operations." In this way, when the agent executes video or 3D generation tasks, it can first perform semantic reasoning-based static image generation, and then invoke video generation or 3D generation tools to achieve semantically rich multimodal outputs.
>
> 3. If there are multiple video generation or 3D generation tools in the tool library, the performance boundary information of the tools can similarly be constructed based on the performance-driven selection modeling method proposed in this paper, enabling a more accurate tool selection process.
>
> **References:**
>
> [1] Wen H, Huang Z, Wang Y, et al. Ouroboros3d: Image-to-3d generation via 3d-aware recursive diffusion[C]//Proceedings of the Computer Vision and Pattern Recognition Conference. 2025: 21631-21641.
>
> [2] Huang Z, Boss M, Vasishta A, et al. Spar3d: Stable point-aware reconstruction of 3d objects from single images[C]//Proceedings of the Computer Vision and Pattern Recognition Conference. 2025: 16860-16870.

---

### Official Review · Reviewer_8HZR · 2025-10-30

**Soundness:** 3
**Presentation:** 4
**Contribution:** 3
**Rating:** 6
**Confidence:** 4

**Summary:**

This paper proposes PerfGuard, a performance-aware agent framework for visual content generation (AIGC). The core idea is to move beyond relying on vague text descriptions for tool selection. Instead, it proposes three mechanisms: (1) PASM, which models tool capabilities using a fine-grained, multi-dimensional scoring matrix ($M_p$); (2) APU, an online feedback mechanism to dynamically update this matrix based on execution outcomes; and (3) CAPO, a training strategy (inspired by DPO/SPO) to fine-tune the agent's planner to align its sub-task decomposition with the known tool capabilities.

The authors claim that this framework leads to SOTA performance on several AIGC benchmarks compared to existing agent-based methods.

**Strengths:**

Novel and Important Problem: The paper correctly identifies a key weakness in current agent frameworks: the "idealized assumption" of tool capabilities. The core idea of modeling fine-grained "performance boundaries" (PASM) is a significant and logical step forward for the field.

Complete Framework: The authors propose a complete, end-to-end vision, including selection (PASM), online adaptation (APU), and planner alignment (CAPO). This holistic approach is commendable.

Clear Presentation: The paper is well-written, and the core concepts are easy to grasp.

**Weaknesses:**

Unfair Comparison & Critical Confounding Variable: This is the most severe flaw. According to Appendix A.2, the framework uses GPT-4o for the Analyst and Self-Evaluator roles. This Self-Evaluator (GPT-4o) provides the entire learning signal for both the CAPO training (via Eq. 4 & 5) and the APU updates (via $R_{actual}$). The Analyst (GPT-4o) is also used during inference. The baseline methods (e.g., GenArtist) are not afforded this powerful, external model. Therefore, the SOTA claims are invalid. The paper is essentially comparing a system distilled from and assisted by GPT-4o against open-source models, which is an unfair comparison.

Lack of Disentanglement (The "1+1>2" Problem): The ablation study in Table 4 fails to answer the most critical question: where does the performance gain come from? Is it the better planner (CAPO) or the better selector (PASM)? The gains might just be additive. A crucial experiment is missing: Smart Planner (CAPO) + Naive Selector (Text-based). Without this, the value of the complex PASM/APU system is unclear, as is the value of the CAPO training.

Practicality and the "Cold-Start" Problem: The paper's claim of "practical utility" is undermined by its reliance on pre-existing benchmarks (T2I-CompBench, ImgEdit-Bench) to create the $M_p$ matrix (Section 4.1). This completely avoids the hardest and most expensive part of the problem: how to build this fine-grained performance matrix from scratch for a new domain or new set of tools. This significant limitation (which I know from experience is a major barrier) is not adequately discussed.

**Questions:**

[Re: Unfair Comparison]: Given the critical dependence on GPT-4o, can the authors please provide an ablation study where the Self-Evaluator and Analyst are replaced with a strong open-source VLM (e.g., LLaVA, Qwen-VL) or a non-LLM metric (e.g., CLIP Score)? This is essential to prove that the performance gains come from the PerfGuard framework itself and not from GPT-4o.

[Re: Disentanglement]: To prove the synergistic value of the framework, can the authors please provide the missing ablation: Baseline + CAPO only (i.e., the smart planner with the original naive, text-based tool selector)? This would allow us to understand the independent contribution of the planner and selector, and to check for any "1+1>2" effect.

[Re: Cold-Start Problem]: Can the authors please elaborate on the scalability of their approach? How would one feasibly construct the $M_p$ matrix in a new domain that lacks a comprehensive, multi-dimensional benchmark like T2I-CompBench? Acknowledging this as a major limitation would be a start.

---

> ### Author Response · Authors · 2025-11-20
> **Response to Reviewer 8HZR (part1)**
>
> **Weaknesses-Question 1: The issue of unfair comparison: The compared baseline uses models that do not have access to powerful external LLMs, whereas our method utilizes such models, resulting in an unfair comparison.**
>
> **Answer 1:** We understand your concerns regarding the fairness of the comparison in this paper and apologize for the shortcomings in the way it was presented. We will provide a detailed explanation of the issues you raised.
>
> 1. In the proposed method, Analyst and Self-Evaluator use GPT-4o primarily to better understand image information. However, compared to GPT-4o, open-source MLLMs have lower performance in image understanding, which limits the ability of Analyst to analyze the semantics of the input image, as well as the ability of Self-Evaluator and APU to score and compare the image. As a result, this somewhat restricts the potential of PerfGuard and limits its task execution performance.
> 2. In fact, among the agent-based baselines we compared, GenArtist is based on GPT-4v, and T2I-Copilot is based on GPT-4o-mini, both of which are closed-source and high-performance MLLMs. In contrast, the Planner and Worker in our method are based on the open-source Qwen3-8B and Qwen3-14B. Therefore, to some extent, the performance of the LLMs involved in task execution planning and tool invocation in our method is lower than that of the models used in the compared methods.
> 3. To further ensure fairness, we replaced the MLLMs in each module (Analyst, Planner, Worker, and Self-Evaluator) of GenArtist, T2I-Copilot, and PerfGuard with Qwen3-VL-32B for comparison. The experimental results are as follows:
> | Method                        | Color  | Spatial | Complex | Degradation (Complex) |
> | ----------------------------- | ------ | ------- | ------- | --------------------- |
> | GenArtist (Qwen3-VL-32B)      | 0.5670 | 0.2928  | 0.2321  | 48.4%                 |
> | GenArtist (Original Config)   | 0.8482 | 0.5437  | 0.4499  | --                    |
> | T2I-Copilot (Qwen3-VL-32B)    | 0.6755 | 0.2257  | 0.2461  | 38.2%                 |
> | T2I-Copilot (Original Config) | 0.8039 | 0.3228  | 0.3985  | --                    |
> | Ours (Qwen3-VL-32B)           | **0.8500** | **0.5481**  | **0.4538**  | **9.4%**                  |
> | Ours (Original Config)        | **0.8753** | **0.6120**  | **0.5007**  | --                    |
>
> The experimental results indicate that: i) Even with Qwen3-VL-38B replacing all LLMs, our method still outperforms existing approaches, demonstrating its superior architecture, tool selection, and task planning efficiency.  ii) Replacing PerfGuard with Qwen3-VL-38B causes only a slight performance drop. Compared to GenArtist (48.4%) and T2I-Copilot (38.2%), PerfGuard's performance decreases by just 9.4%, demonstrating its better adaptability and robustness across different LLMs. The details and discussion of this content are presented in Supplementary materials **A.5** of the revised paper.

---

> ### Author Response · Authors · 2025-11-20
> **Response to Reviewer 8HZR (part2)**
>
> **Weaknesses-Question 2: The ablation study lacks a comparison of the performance between CAPO+ Text-based performance description tool selection.**
>
> **Answer 2: **We sincerely apologize for the shortcomings in the ablation study that led to your confusion. To provide a clearer understanding of the performance comparison of the PerfGuard modules and clarify the contribution of each module to the final results, we have added a performance comparison with CAPO+ and text-based tool selection. The experimental results are as follows:
>
> | CAPO | PASM | APU  | Color ↑ | Spatial ↑ | Complex ↑ |
> | ---- | ---- | ---- | ------- | --------- | --------- |
> | ✕    | ✕    | ✕    | 0.8239  | 0.5600    | 0.4327    |
> | ✓    | ✕    | ✕    | 0.8466  | 0.5756    | 0.4493    |
> | ✕    | ✕    | ✓    | 0.8521  | 0.5919    | 0.4412    |
> | ✕    | ✓    | ✓    | 0.8596  | 0.6005    | 0.4738    |
> | ✓    | ✓    | ✓    | **0.8753**  | **0.6120**    | **0.5007**    |
>
> The experimental results clearly show that:
> - Relying solely on conventional text descriptions for tool capabilities often leads to misselection, forcing the Worker to perform near-exhaustive attempts, resulting in the lowest performance.
> - Even with the Capability-Aligned Planning Optimization mechanism, the lack of an accurate tool selection strategy hinders the Planner's task planning, resulting in limited overall performance improvement.
> - Introducing the Performance-Aware Selection Modeling  mechanism significantly improves some metrics, with the color dimension increasing by 3.42\% and the texture dimension by 5.7\%.
> - Further applying Adaptive Preference Updating fine-tunes preference scores for Planner-generated sub-tasks, enhancing tool selection precision and raising the complex dimension from 0.4412 to 0.4738.
>
> - The performance-driven tool selection strategy improves tool selection accuracy in downstream tasks, enhancing sub-task execution efficiency. This, in turn, boosts overall task planning accuracy by the Planner, further optimizing PerfGuard’s performance with CPAO support.
>
> The details and discussion of this content are presented in **Section 5.3** (Ablation Study) of the revised paper.

---

> ### Author Response · Authors · 2025-11-20
> **Response to Reviewer 8HZR (part3)**
>
> **Weaknesses-Question 3: The issues of over-reliance on existing benchmarks, the cost of constructing the performance boundary matrix, and the scalability of the proposed method.**
>
> **Answer**: We appreciate the reviewer’s concerns about the scalability and cost of our method. Here is our explanation:
>
> 1. **Reliance on Existing Benchmarks**: The benchmarks used in our method serve only to initialize the performance boundary matrix. Like reinforcement learning, our approach iteratively updates the matrix based on the actual execution performance of sampled tools (APU), internalizing tool selection experience. This allows for quick adaptation to any tool without requiring additional LLM parameter adjustments. Benchmarks like T2I-CompBench are authoritative tools for image generation and provide highly accurate evaluations. Even without high-quality benchmarks in certain fields, PerfGuard can update the matrix based on the actual performance of the tools in use. Therefore, our method does not overly rely on existing benchmarks.
> 2. **Cost of Performance Boundary Matrix Construction**: PerfGuard allows users to define the dimensions of the task results they care about. The matrix scores are updated during tool execution by comparing the performance of different tools on the task, based on LLM feedback or user input. This is a low-cost, ongoing evaluation and optimization process, in contrast to the high cost of data collection or trajectory experience definition in current LLM-based methods. Thus, the matrix construction is more efficient and cost-effective.
> 3. **Scalability to New Domains**: The method is also applicable to new domains, with the following steps:
>    - **(1)** Users define the relevant dimensions for the new domain, specifying their meaning in the worker's prompt.
>    - **(2)** The performance boundary matrix for new tool libraries can be initialized using an average or random initialization, allowing different tools to be selected during the performance score update.
>    - **(3)** The APU process updates the matrix iteratively by comparing the execution of various tools, ensuring performance updates based on user preferences.

---

### Official Review · Reviewer_t5Fu · 2025-11-01

**Soundness:** 3
**Presentation:** 3
**Contribution:** 2
**Rating:** 6
**Confidence:** 4

**Summary:**

The paper proposes PerfGuard, a performance-aware agent framework for visual content generation. It aims to address the gap in current agent-based systems, which often operate under the assumption that tool executions are always successful. PerfGuard incorporates three core mechanisms: Performance-Aware Selection Modeling (PASM), Adaptive Preference Update (APU), and Capability-Aligned Planning Optimization (CAPO). These mechanisms model tool performance boundaries, optimize tool selection, and align task planning with execution outcomes, providing enhanced reliability in visual content generation tasks.

**Strengths:**

1. The framework introduces a novel approach to visual content generation by incorporating performance-aware mechanisms.

2. The introduction of a multi-dimensional performance evaluation system for tools is a clear strength. It provides a more detailed and reliable method for tool selection, addressing the limitations of previous systems that relied on general textual descriptions.

**Weaknesses:**

1. The method heavily relies on the context-learning capabilities of large language models (LLMs). While this is an interesting approach, it may not offer substantial improvements over previous systems that already utilize LLMs for similar tasks.

2. The paper does not sufficiently discuss existing methods, particularly in relation to visual content editing tools. For example, "CLOVA: A Closed-Loop Visual Assistant with Tool Usage and Update" (CVPR 2024) has already considered tool evaluation and updates. This overlap should be addressed in the discussion to strengthen the paper's contribution.

3. While LLMs have shown remarkable capabilities in various domains, their use in image editing tool evaluation and updates is still in the early stages. The paper should address the potential limitations and challenges of this approach, including issues related to LLM performance, tool generalization, and robustness in diverse image editing tasks. The authors should try different LLMs to show the robustness of this method.

**Questions:**

see the weaknesses

---

> ### Author Response · Authors · 2025-11-20
> **Response to Reviewer t5Fu (part1)**
>
> **Weaknesses 1: The proposed method heavily relies on large models and does not offer substantial improvements compared to existing LLM applications.**
>
> **Answer 1:**  We understand your concerns regarding the method we proposed:
>
> 1. Traditional agent-based methods relying solely on LLM context-learning struggle with tool selection due to two issues: 1) textual descriptions often fail to capture tool capabilities accurately, especially for similar tools with different performance preferences, and 2) scaling becomes inefficient as tool libraries grow, since LLMs face memory limits and hallucination issues when processing long descriptions.
>
> 2. We need to clarify that our method does not heavily rely on the context-learning capabilities of LLMs. Instead, it focuses on accurate tool-task matching through performance-aware selection, addressing the inefficiencies of context-based methods. The primary issue we address is the inefficiency and inaccuracy of tool invocation in previous methods that solely rely on context-learning, a problem not fully considered in prior research.
>
> 4. In the new experiment 5.4, we compared token consumption between traditional context-learning-based tool selection and PerfGuard in a large-scale tool library. The experimental results are as follows:
>
> | Method/Tool Num | 10 | 40 | 70 | 100 | 130 | 160 | 190 | 200 |
> | :--- | :--- | :--- | :--- | :--- | :--- | :--- | :--- | :--- |
> | Context-learning Method | 1469.3 | 3002.9 | 4489.6 | 6030.3 | 7630.3 | 9139.3 | 10802.6 | 11308.7 |
> | Ours (d=4) | 570.8 | 587.0 | 584.3 | 578.8 | 593.5 | 576.9 | 587.6 | 589.3 |
> | Ours (d=8) | 755.0 | 768.8 | 741.2 | 757.6 | 751.5 | 736.8 | 716.4 | 754.1 |
> | Ours (d=12) | 846.9 | 865.0 | 835.4 | 827.5 | 845.0 | 852.8 | 851.3 | 863.5 |
> | Ours (d=16) | 981.1 | 977.8 | 982.6 | 1028.9 | 1012.3 | 988.1 | 986.5 | 967.9 |
>
>    The results show that while token consumption of the context-learning-based method grows catastrophically as the number of tools increases (10–200), PerfGuard remains low and stable. This demonstrates that our method does not heavily rely on the context-learning capabilities of LLMs. On the contrary, the problem we address is the limitations of context-learning-based methods in tool selection.
>
> ---
>
> **Weaknesses 2:  It is necessary to consider and add a discussion on similar work, "CLOVA: A Closed-Loop Visual Assistant with Tool Usage and Update (CVPR 2024)."**
>
> **Answer 2:**  Thank you for highlighting the shortcomings of our work. We have addressed these concerns in the revised paper by adding a discussion in the "Related Work" section to clarify our method's contributions and advantages. While we acknowledge similarities with "CLOVA," such as the use of reflection and evaluation for tool updates, our approach differs significantly in the problems addressed and the methodologies used:
>
> 1. **Differences in problem and approach**: Our method focuses on tool evaluation and updates to select the most suitable tool based on execution performance, improving task efficiency. In contrast, "CLOVA" enhances tool success rates by refining the prompt pool through self-reflection and prompt tuning.
> 2. **Different perspectives on the problem**: Existing research mainly addresses improving tool performance via self-reflection and prompt tuning, focusing on using tools correctly. However, the challenge of selecting the right tool, especially among those with similar capabilities, has been largely overlooked. Our approach solves this by establishing performance boundaries and an iterative update mechanism.
>
> We believe that this work (CLOVA) makes an important contribution to the Agent field and has provided us with valuable insights. In addition to incorporating a discussion of this work in our paper, we will also explore its research ideas in our future studies to address other related issues.

---

> ### Author Response · Authors · 2025-11-20
> **Response to Reviewer t5Fu (part2)**
>
> **Weaknesses 3: This paper needs to discuss the potential limitations and challenges of the proposed method and attempt to validate its robustness through the use of different LLMs.**
>
> **Answer 3:** Thank you for pointing out the insufficient discussion of the limitations and robustness of the proposed method in this paper. We will address your concerns in the following points, and the corresponding content will be added and modified in the sections marked in blue in the paper:
>
> 1. PerfGuard addresses the tool selection and capability matching problem in image generation and editing, a challenge not previously explored in LLM research. In **Experiment 5.4** (Efficiency Comparison), we show that the text-based selection method is inefficient for large tool libraries, while our approach ensures computational costs remain stable, even as the library grows, distinguishing it from prior methods.
>
> 2. PerfGuard is an initial attempt to solve accurate tool selection in agent systems, but it has limitations. The Performance-Aware Selection strategy uses benchmark scores to initialize the tool performance matrix. However, in domains outside visual content generation, high-quality benchmarks may not be available for initialization. Future work will extend this method to other domains, like visual reasoning, and enhance multi-agent collaboration. We’ve added a section on limitations and challenges in **Sec. 6** of the revised paper.
>
> 3. We validated PerfGuard's generalization and robustness by replacing the MLLM with GPT-4o (closed-source) or Qwen3-VL-38B (open-source). Due to GPT-4o's closed nature, we used an experience replay mechanism to match PerfGuard’s performance with CAPO’s planning accuracy. The experimental results are as follows:
>
> | Method                        | Color  | Spatial | Complex | Degradation (Complex) |
> | ----------------------------- | ------ | ------- | ------- | --------------------- |
> | GenArtist (Qwen3-VL-32B)      | 0.5670 | 0.2928  | 0.2321  | 48.4%                 |
> | GenArtist (Original Config)   | 0.8482 | 0.5437  | 0.4499  | --                    |
> | T2I-Copilot (Qwen3-VL-32B)    | 0.6755 | 0.2257  | 0.2461  | 38.2%                 |
> | T2I-Copilot (Original Config) | 0.8039 | 0.3228  | 0.3985  | --                    |
> | Ours (Qwen3-VL-32B)           | 0.8500 | 0.5481  | 0.4538  | **9.4%**                    |
> | Ours (GPT-4o)                 | 0.8577 | 0.6004  | 0.4813  | **3.9%**                  |
> | Ours (Original Config)        | 0.8753 | 0.6120  | 0.5007  | --                    |
>
> Replacing PerfGuard with Qwen3-VL-38B or GPT-4o causes only slight performance degradation. Compared to GenArtist (48.4%) and T2I-Copilot (38.2%), PerfGuard shows minimal drop (9.4% and 3.9%) in the Complex dimension of T2I-CompBench, highlighting its superior adaptability across both open- and closed-source LLMs and confirming its robustness and generalization.

---

### Author Response · Authors · 2025-11-26
**Global Responses to all Reviewers and AC**

We sincerely thank the reviewers for their time and valuable feedback. Their suggestions have significantly improved the quality of our work. The reviewers consistently recognized our method's innovative performance-aware fine-grained tool selection mechanism, which effectively resolves the ambiguity in tool capability boundaries in text descriptions. Detailed comparisons in image generation and editing tasks have validated the effectiveness of the proposed PerfGuard.

Below, we summarize key clarifications and changes made in response to the reviewers' feedback:

### **Key Clarifications:**
- **Main Contribution**: Our primary contribution is a novel tool capability boundary modeling and selection mechanism. Unlike most LLM-based studies, which overlook the limitations of text descriptions when tool capabilities are similar or when the tool count increases, our method uses multidimensional evaluation scores to represent tool preferences, achieving a fine-grained match between task requirements and tool capabilities.

- **Advantages over Context-Based LLM Tool Selection**: Traditional context-based methods require inputting detailed descriptions of all tools into the context, which increases error rates and token consumption as the tool library grows. Our Performance-Aware Selection Modeling addresses these issues, significantly improving tool selection accuracy and efficiency, with minimal impact on token consumption regardless of tool count.

- **Scalability**: Our method is not tied to any specific tool type and is compatible with a wide range of tools. For fairness, we focused on text-based generative and editing tools in our experiments. While benchmarks accelerate the convergence of Adaptive Preference Updating, our method can still update the tool performance boundary matrix even without high-quality benchmarks.

### **Revisions and Additions in the Updated Paper:**
- **Ablation Experiment** (Sec. 5.3): We added a comparison between CAPO and text-based tool selection metrics (Table 4) to demonstrate the performance impact of training the Planner alone and the combined performance gains with Performance-Aware Selection Modeling and CAPO.

- **Token Consumption Comparison** (Sec. 5.4, Figure 8): A new simulation experiment compares token consumption for text-based tool descriptions and our method as the number of tools increases, highlighting the efficiency differences in large tool libraries.

- **Time Consumption Comparison** (Sec. 5.4, Figure 7): We added a comparison of time consumption during inference between our method and baseline agents, emphasizing our method's time efficiency advantages.

- **User Study** (Appendix A.3): A user study with visualizations (Figure 9) and both subjective and objective evaluations (Table 5) compares the user experience and performance of our method versus baselines.

- **Performance Comparisons with Same LLM** (Appendix A.5, Table 6): We added performance comparisons to verify the advantages of our method when LLM performance is controlled.

- **Ablation Experiments with Different LLMs**: We incorporated ablation experiments using both open-source and closed-source LLMs to verify the robustness of our method across different LLMs.

- **Related Work**: We extended the related work section with a discussion of the "CLOVA" method to clarify the contributions and advantages of our approach.

- **Limitations and Challenges** (Sec. 6): We included a "Limitations and Challenges" section to discuss the method’s limitations and provide future research directions.

All of the modifications mentioned above are highlighted in **blue text** within the paper.

We sincerely appreciate the reviewers' detailed feedback, which has significantly strengthened our paper. We hope these revisions address the concerns raised and enhance the clarity and robustness of our work. Thank you again for your time and consideration.

---

### Meta-Review · Area_Chair_CLrY · 2025-12-31

**Summary:**

This paper proposes PerfGuard, a performance-aware agent framework for visual content generation. All reviewers acknowledge the novelty and the completeness of the proposed framework with thorough experimental results. Besides, reviewers also point out some concerns for this paper:

1. Moderate improvements
2. Related works
3. Comparisons and experiments on more LLMs and baselines

Authors have provided massive response to address reviewers' concerns. Considering the positive feedback from all reviewers, I recommend acceptance for this paper.

**Reviewer Concerns:**

Authors have provided more results to demonstrate their method take few token computations compared with other methods, which manifest its efficiency. Besides, they also provided more discussion about related works, and results on new LLMs, and also user studies. These responses can be useful to address reviewers' concern.

**Reviewer Scores:**

Based on authors' response, reviewer can maintain or increase their scores.

---

### Decision · Program_Chairs · 2026-01-26

Accept (Poster)